# EG3AD: An Efficient Geometry-Aware Encoding Framework for Reconstruction-Based Multi-Class Point Cloud Anomaly Detection

## Abstract

Multi-class point cloud anomaly detection is a critical task that aims to identify anomalous patterns across various categories using a single, unified model. Current reconstruction based methods predominantly rely on transformer encoders to extract high-level semantic features, aiming to filter out subtle defect features, and then use decoders to reconstruct them into normal patterns. However, this suffers from two limitations: (1) employing encoders based on global attention mechanisms, particularly on uniformly tokenized inputs, hinders the rapid extraction of fine-grained local features; (2) high computational cost arising from stacking multiple encoding layers during semantic feature extraction. Thus, we propose EG3AD, an Efficient Geometry-aware encoding framework for reconstruction-based multi-class 3D point cloud Anomaly Detection. To investigate how to obtain effective geometric representations under token and parameter constraints, we begin by introducing the Curvature-Aware Sampling module, which mitigates the distortions caused by uniform sampling in regions of high curvature. Then, leveraging geometry prior bias of point cloud data, we design the Point Cluster Graph Convolution, which enables compact and effect local geometric aggregation through only limited lightweight layers. Finally, to obtain anomaly-invariant semantic features without relying on deep encoding layers, we introduce the Feature Purification Module inspired by optimal transport theory. This module compresses features into a set of cluster centroids that preserve fundamental geometric patterns, thereby yielding representations robust to subtle anomalies. Extensive experiments show that simply replacing the vanilla point transformer encoder with our proposed EG3AD yields state-of-the-art results on all PCAD benchmarks. Our code will be made publicly available upon acceptance.

## 1 INTRODUCTION

Point Cloud Anomaly Detection (PCAD) has attracted growing attention. It aims to identify and localize anomalous points or regions within point cloud data, and can effectively address the problem where 2D image-based anomaly detection methods (Luo et al., 2025; Dai et al., 2024; Fang et al.) struggle with detecting anomalies that involve subtle geometric deformations without prominent texture cues (Horwitz & Hoshen, 2023). Recently, due to the inability of single-class-one-model methods (Figure 1 a) to adapt to diverse production lines, which hinders practical applications, some methods have emerged that use large pretrained networks combined with anomaly detection frameworks to enable a single model to detect multiple categories simultaneously (Figure 1 b). However, these multi-class PCAD methods face several issues:

First, the pre-trained models they rely on (e.g., Transformer-based architectures) (Cheng et al., 2025a; Li et al., 2024) incur quadratic computational complexity to the number of input tokens, which becomes a bottleneck when a large number of tokens are required to represent complex geometries. In terms of the number of parameters, as shown in Figure 2 a and b, PCAD methods based on pretrained model such as Point-MAE (Pang et al., 2022)—which incorporates standard Transformers—require over 20 million parameters, and their complexity increases rapidly with the number of input points. However, practical PCAD applications often operate under strict constraints

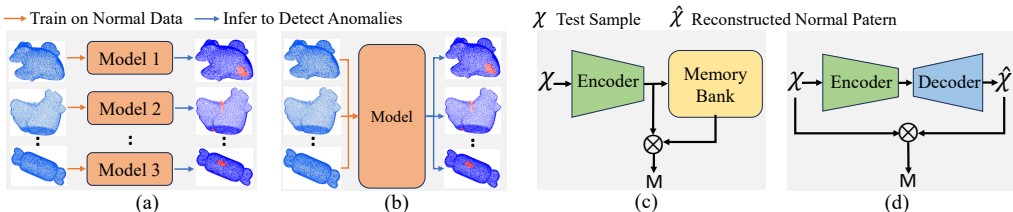

Figure 1: (a) The single-class-one-model paradigm. (b) The multi-class-one-model paradigm. (c) Embedding-based approach. (d) Reconstruction-based approach.

on storage and computational cost (Stropeni et al., 2025). In such scenarios, lightweight networks like PointNet++ (Qi et al., 2017b) are preferred due to their lower parameter count (only 1.5M).

Second, to obtain anomaly-invariant features, deep networks are often employed, either by stacking multiple encoding layers (Cheng et al., 2025a; Li et al., 2024) or by introducing external learnable compression modules to capture multi-scale representations (Luo et al., 2025). However, both approaches result in heavy parameter overhead. Moreover, the absence of low-level geometric features causes the loss of useful information for high-resolution reconstruction, which may in turn require additional decoder parameters to achieve acceptable results, further increasing computational cost.

To tackle above issues, we begin with introducing a Curvature-Aware Sampling (CAS) strategy. Unlike uniform patch division, CAS actively senses surface curvature and adaptively allocates denser patches in geometrically complex regions, thereby making the "information density" of each patch more uniform, thus enabling the same geometry to be expressed with fewer tokens. Second, to enable more efficient local feature extraction, we propose Point Cluster Graph Convolution (PCGC). Guided by patch-center graphs, PCGC starts from a few initial nodes and progressively expands its receptive field into the neighborhood, aggregating fine-grained local geometric context within a small number of hierarchical layers. By replacing the quadratic-cost global attention in transformers with locality-aware aggregation, PCGC exploits geometric priors of point clouds, thereby reducing computational complexity and the number of parameters. Finally, instead of relying on deeper encoders to aggregate anomaly-invariant global semantic features, we propose the Feature Purification Module (FPM). FPM employs only a single linear layer and leverages optimal transport theory with clustering to compress and denoise local geometric anomaly features, thereby efficiently providing anomaly-invariant guidance for reconstruction. Together, CAS, PCGC, and FPM constitute our EG3AD framework, which is more tailored to the characteristics of reconstruction based PCAD task compared with point transformer encoders.

In summary, (1) EG3AD integrates CAS and PCGC to make the model more efficient and effective, enabling lightweight aggregation of local geometric features through adaptive patch sampling and hierarchical aggregation; (2) it incorporates the FPM to purify features corrupted by local anomalies, thereby avoiding reliance on stacking deep encoder layers to obtain features robust to anomalies; and (3) our method establishes new state-of-the-art results on major PCAD benchmarks and achieves top performance on cross-class balance metrics, demonstrating strong generalization capability.

## 2 RELATED WORK

Early research (Cao et al., 2024; Delitzas et al., 2024; Wang et al., 2023) combined 3D geometry with 2D image features for PCAD. With the release of 3D-specific datasets like Real3D-AD (Liu et al., 2023) and Anomaly-ShapeNet (Li et al., 2024), attention has shifted to detecting anomalies directly in 3D point clouds. PCAD methods can be categorized along two dimensions: by model scope, into single-class-one-model paradigms and multi-class-one-model paradigms; by detection strategy, into embedding-based approaches and reconstruction-based approaches.

### 2.1 SINGLE-CLASS AND MULTI-CLASS PCAD

Initial methods typically followed a single-class-one model(Figure 1 a) learning paradigm. Representative approaches include embedding-based methods such as BTF (Horwitz & Hoshen, 2023) (using FPFH), M3DM (Wang et al., 2023) (based on PointMAE), CPMF (Cao et al., 2024), Reg3D-

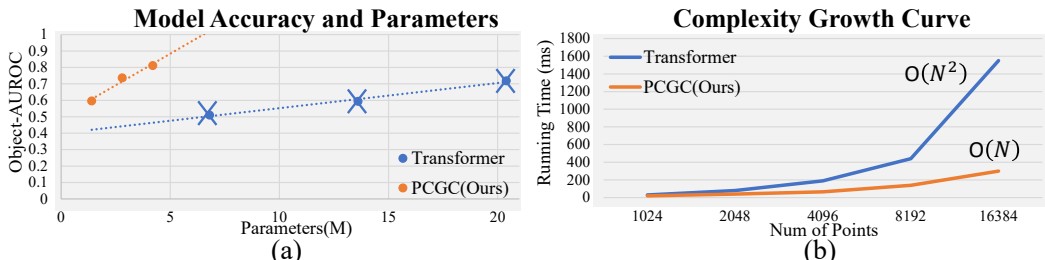

Figure 2: Comparison of our PCGC and Transformer in terms of performance and efficiency.

AD (Liu et al., 2023), Group3AD (Zhu et al., 2024) and PO3AD (Ye et al., 2024). Among these methods, PO3AD, which is based on reconstruction models, achieves promising results in the single-class setting. However, all these methods require a separate model for each category, which hinders their scalability and practical deployment.

Recent interest in multi-class PCAD(Figure 1 b) has led to methods leveraging reconstruction models with strong generalization, such as IMRNet (Li et al., 2024), GLFM Cheng et al. (2025b) and R3D-AD (Zhou et al., 2024), which build on Point-MAE or diffusion backbones. MC3D-AD (Cheng et al., 2025a) further improves performance by refining the training strategy of Point-MAE. However, these models rely on backbones designed for generic tasks and lack the capacity to rapidly learn fine-grained geometric knowledge from limited PCAD data, which motivates the design of our EG3AD framework.

### 2.2 EMBEDDING AND RECONSTRUCTION BASED PCAD

Feature embedding-based methods(Figure 1 c)Horwitz & Hoshen (2023); Wang et al. (2023); Cao et al. (2024); Chu et al. (2023), typically encode input $X$ into latent features stored in a memory bank, generating anomaly maps $M$ by comparing test features with stored ones. Reconstruction-based methods(Figure 1 d) (Zhou et al., 2024; Li et al., 2024; Cheng et al., 2025a) are trained to reproduce normal point clouds, so that when presented with inputs that may contain anomalies at test time, the model attempts to reconstruct their normal counterparts. The anomaly map $M$ is then obtained from the discrepancy between the input $X$ and its reconstruction $\hat{X}$. These studies demonstrate that reconstruction-based models generalize well (Liu et al., 2024; Lin et al., 2025), making them a promising solution for multi-class PCAD.

## 3 METHODOLOGY

### 3.1 OVERVIEW

As illustrated in Figure 3, EG3AD adopts a reconstruction-based paradigm, where the objective is to restore anomalous samples containing subtle defects into their corresponding normal counterparts. The anomalies are then detected by comparing the reconstructed outputs with the original inputs. The core innovations of EG3AD lie in the CAS, PCGC, and FPM. And following a curriculum-inspired training strategy, the model achieves remarkable performance even when trained with only a limited number of normal samples.

### 3.2 CURVATURE-AWARE SAMPLING

CAS is a strategy that biases sampling toward high-curvature regions to improve representational fidelity without increasing computational cost. As illustrated in Figure 3 a, it begins by oversampling the input point cloud $P \in \mathbb{R}^{n \times 3}$ using Farthest Point Sampling (FPS) (Qi et al., 2017a) to generate $m' > m$ centroids, where $m$ denotes the final target number of patches. For each centroid $p_i$, a local patch is formed using its $k$-nearest neighbors. Specifically,

$$P_{\text{initial}} = \text{FPS}(P), \tag{1}$$

$$C_{\text{initial}} = \{\text{kNN}(P, p_i, k) \mid p_i \in P_{\text{initial}}\}, \tag{2}$$

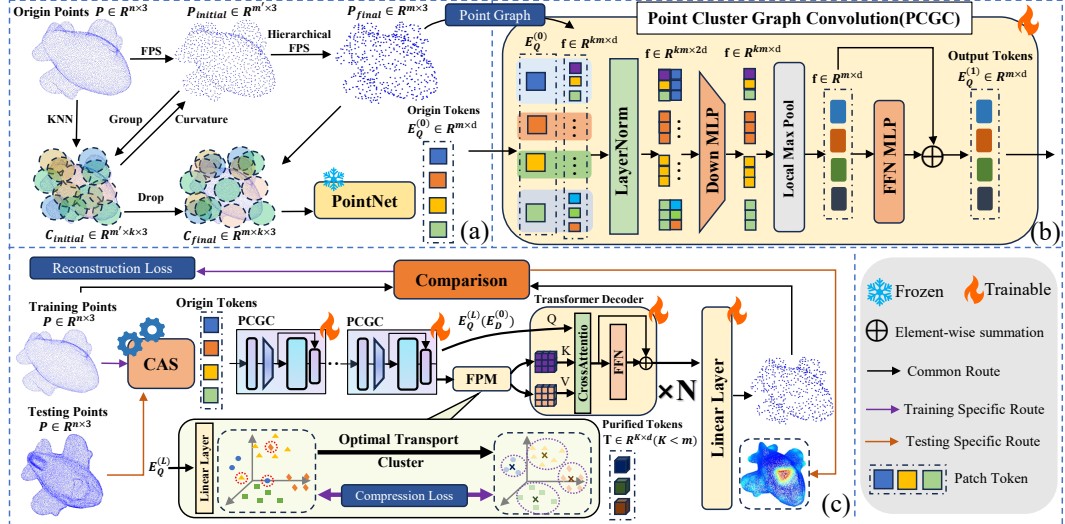

Figure 3: An overview of our EG3AD framework. The model is composed of a CAS module, three PCGC encoder layers, a FPM, and reconstruction decoder layers guided by the compressed features. (a) First, the CAS module adaptively samples the input point cloud and encodes the resulting patches into tokens. (b) Next, multiple PCGC layers efficiently extract fine-grained local features by operating on the point cluster graph. (c) The FPM then compresses these features, and the decoder leverages the compressed representation to reconstruct an anomaly-free sample.

Then, with a tiered ranking function $f_{CT}$, a mean curvature score is computed for each patch to quantify its geometric complexity. Based on these scores, the patches are categorized into three tiers: high-curvature ($P_{\text{high}}$), medium-curvature ($P_{\text{mid}}$), and low-curvature ($P_{\text{low}}$). In order to obtain $m$ final patches, ($m' - m$) patches are discarded by applying a higher drop rate to the low-curvature tier. This results in three retained subsets: $P_{\text{low}} \in \mathbb{R}^{r_1 m \times 3}$, $P_{\text{mid}} \in \mathbb{R}^{r_2 m \times 3}$, and $P_{\text{high}} \in \mathbb{R}^{r_3 m \times 3}$, where $r_3 > r_2 > r_1$. This selective retention ensures that the final set of patches densely covers geometrically intricate regions. The overall procedure is formally summarized as:

$$(P_{\text{high}}, P_{\text{mid}}, P_{\text{low}}) = f_{CT}(C_{\text{initial}}), \tag{3}$$

$$P_{\text{final}} = \text{FPS}(P_{\text{high}}) \cup \text{FPS}(P_{\text{mid}}) \cup \text{FPS}(P_{\text{low}}), \tag{4}$$

$$C_{\text{final}} = \{ c \in C_{\text{initial}} \mid \text{center}(c) \in P_{\text{final}} \}. \tag{5}$$

By enforcing a higher retention rate for high-curvature patches (i.e., preserving a larger fraction of $P_{\text{high}}$), this adaptive sampling mechanism provides a strong geometric foundation for detecting subtle anomalies, in contrast to conventional FPS which treats all regions uniformly.

## 3.3 POINT CLUSTER GRAPH CONVOLUTION

Transformer based models, shine when there is abundant training data and sufficient depth/width, since their global self-attention can capture long-range dependencies but lacks strong built-in inductive bias. Convolutional networks encode translation equivariance and local priors by design, which makes them more data-efficient and better when network capacity is limited. (Dosovitskiy et al., 2020). Therefore, in the context of PCAD, we propose PCGC to efficiently model the fine-grained geometric details of normal samples.

The PCGC module (Figure 3 b) first constructs a geometric graph $\mathscr{G}$ based on $C_{\text{final}}$, where each cluster is treated as a node and edges are established between geometrically adjacent clusters. Let $E_Q^{(l-1)}$ denote the input to the $l$-th PCGC layer, specifically,

$$E_Q^{(0)} = \text{PointNet}(C_{final}), \tag{6}$$

$$\mathscr{G} = \{\text{kNN}(P_{\text{final}}, \boldsymbol{p}_i, k) \mid \boldsymbol{p}_i \in P_{\text{final}}\}. \tag{7}$$

Rather than employing global attention to aggregate information across all patch tokens, PCGC performs localized feature aggregation. Similar to a Convolutional Neural Network (CNN) (Long et al., 2015), each cluster aggregates features solely from its immediate neighbors within the graph. By stacking multiple PCGC layers, the effective receptive field of each cluster progressively expands, enabling the model to capture features from local to semi-global contexts in a controlled and hierarchical manner. The forward computation at layer $l$ is defined as:

$$\boldsymbol{E}_Q^{(l)} = f_{\text{PCGC}}(\boldsymbol{E}_Q^{(l-1)}, \mathscr{G}) \quad \text{for } l = 1, 2, \ldots, n. \tag{8}$$

In each PCGC block, the $k$-nearest neighbors algorithm is first applied based on the geometric graph $\mathscr{G}$ to identify the $k$ nearest neighbor features $\mathscr{G}(\boldsymbol{E}_Q^{(l-1)}) \in \mathbb{R}^{kn \times d}$ for each token in the input feature map $\boldsymbol{E}_Q^{(l-1)}$. Each token is then replicated $k$ times and concatenated with its corresponding neighbors to form an enriched representation in $\mathbb{R}^{kn \times 2d}$. A shared MLP subsequently performs a nonlinear transformation over all local pairs to extract fine-grained geometric patterns. Finally, a max pooling operation aggregates the transformed local features to produce the output for each patch.

Each block contains only a single simple MLP layer, significantly reducing the number of network parameters. Additionally, since static local point graph only need to be computed once at the beginning and can be shared across all layers, the design avoids not only the dynamic attention computations required by transformer-based models in each layer, but also the repeated sampling and grouping of features in every layer of PointNet++, thereby reducing the computational overhead.

### 3.4 Feature Purification Module

PCGCs extract local geometric features; however, to enable the decoder to reconstruct normal patterns from these features while incorporating defect information, an anomaly-invariant reference is required. Previous approaches typically address this by stacking multiple encoder layers to obtain deep semantic features, which can filter out local geometric anomalies. Nevertheless, such designs introduce a large number of parameters, resulting in computational inefficiency. To overcome this limitation, we propose the FPM, which leverages optimal transport-based clustering to construct the anomaly-invariant reference. Specifically, FPM compresses the encoded point features into a compact set of structural tokens under an equipartition constraint, drawing inspiration from optimal transport theory. As illustrated in Figure 3 c, the module takes as input the feature map $\boldsymbol{E}_Q^{(L)} \in \mathbb{R}^{m \times d}$ from the final encoder layer.

The objective is to compress $\boldsymbol{E}_Q^{(L)}$ into a fixed-size set of $K$ cluster summary tokens $\boldsymbol{T} = \{\mathbf{t}_k\}_{k=1}^K \in \mathbb{R}^{K \times d}$, where $K \ll m$. This process unfolds in four sequential steps.

First, we introduce a lightweight clustering head, realized through a linear projection denoted as $h(\cdot)$, which is responsible for computing an assignment logits matrix. This component provides a compact yet effective mechanism for mapping feature representations into cluster-specific scores:

$$\boldsymbol{A} = h(\mathbf{E}_{\mathbf{Q}}^{(\mathbf{L})}), \quad \mathbf{A} \in \mathbb{R}^{m \times K}. \tag{9}$$

Here, each element $\boldsymbol{A}_{i,k}$ signifies the unnormalized affinity of point $i$ to cluster $k$.

Second, a row-wise softmax operation is applied to $\boldsymbol{A}$ to obtain a initial soft assignment matrix $\boldsymbol{P}$:

$$\boldsymbol{P}_{i,k} = \frac{\exp(\boldsymbol{A}_{i,k})}{\sum_{j=1}^K \exp(\boldsymbol{A}_{i,j})}. \tag{10}$$

This ensures each point's assignment probabilities sum to one, satisfying a row-wise normalization constraint.

Third, to enforce a critical equipartition constraint—that each cluster summarizes a roughly equal proportion of points—we employ the Sinkhorn-Knopp algorithm. This optimal transport technique finds a matrix $\boldsymbol{Q}$ that approximates $\boldsymbol{P}$ while satisfying marginal constraints:

$$\boldsymbol{Q} \in \mathbb{R}^{m \times K}: \quad \boldsymbol{Q}\mathbf{1} = \frac{1}{m}\mathbf{1}_m, \quad \boldsymbol{Q}^\top \mathbf{1} = \frac{1}{m}\mathbf{1}_m. \tag{11}$$

The second constraint, $Q^\top 1$, is the equipartition condition, guaranteeing a uniform distribution of probability mass across clusters and preventing collapse.

Finally, a hard assignment is derived for feature aggregation. The cluster label for each point is obtained via $\hat{q}_i = \arg\max_k \mathbf{Q}_{i,k}$. The summary token for the $k$-th cluster is then computed as the average of features from all points assigned to it:

$$\mathbf{t}_k = \frac{1}{|\{i : \hat{q}_i = k\}|} \sum_{i:\hat{q}_i=k} \mathbf{f}_i. \tag{12}$$

The resultant tokens, $T$, constitute a compressed and purified representation of the test sample. $T$ serve as anomaly-invariant reconstruction guidance for the decoder, functioning as the keys and values in the cross-attention of the transformer decoder. A brief illustration from the perspective of mutual information theory is provided in the appendix.

### 3.5 TRAINING OBJECTIVE

The learning of the entire network is governed by two complementary loss functions that work in concert. $\mathscr{L}_{\text{FPM}}$ ensures the cluster head's predictions $P$ align with the equipartitioned, optimal assignment $Q$, enforcing the emergence of $K$ distinct and meaningful clusters. The equipartition constraint protects the representation of smaller, critical structures by guaranteeing their independent representation in the token set $Q$. $\mathscr{L}_r$ measures how well the decoder reconstructs the original encoded features from the compressed ones. Let $\mathbf{e}_{Q_i}^{(0)}$ and $\mathbf{e}_{D_j}^{(L)}$ represent the $i$-th token of the input to the 1st layer's Encoder and the $j$-th token of the output of the last layer's Decoder, respectively. The training objective can be mathematically expressed as:

$$\mathscr{L}_{\text{FPM}} = -\frac{1}{m} \sum_{i=1}^{m} \sum_{k=1}^{K} \mathbf{Q}_{i,k} \log \mathbf{P}_{i,k}, \tag{13}$$

$$\mathscr{L}_r = \frac{1}{m} \sum_{i=1}^{m} \min_j \left\| \mathbf{e}_{Q_i}^{(0)} - \mathbf{e}_{D_j}^{(L)} \right\|_2^2 + \frac{1}{m} \sum_{j=1}^{m} \min_i \left\| \mathbf{e}_{D_j}^{(L)} - \mathbf{e}_{Q_i}^{(0)} \right\|_2^2. \tag{14}$$

This dual-objective design inherently attenuates anomalies, which typically manifest as localized deviations, by leveraging the consistency enforced within each semantically coherent cluster. As a result, anomalous signals are naturally downweighted during the feature aggregation process, while normal patterns are more effectively reinforced.

### 3.6 ANOMALY SCORE COMPUTATION

Given a predicted anomalous point cloud $\tilde{P}$ derived from a normal reference point cloud $P$, we compute the anomaly score for each point $\tilde{p} \in \tilde{P}$ based on its geometric deviation from $P$. Specifically, for each $\tilde{p}$, we first identify its three nearest neighbors $p_1, p_2, p_3 \in P$ and compute the average Euclidean distance:

$$\tilde{l}(\tilde{p}) = \frac{1}{3} \sum_{i=1}^{3} \|\tilde{p} - p_i\|_2. \tag{15}$$

We then normalize these distances to obtain the final anomaly score $s(\tilde{p})$. However, instead of linearly mapping $\tilde{l}(\tilde{p})$ into $[0, 1]$, we adopt a dynamic scaling strategy to prevent low-deviation samples from being spuriously mapped near 1. The anomaly score is defined as:

$$s(\tilde{p}) = \frac{\tilde{l}(\tilde{p}) - \tilde{l}_{\min}}{\tilde{l}_{\max} - \tilde{l}_{\min} + \varepsilon(\tilde{l}_{\max})}, \tag{16}$$

where $\tilde{l}_{\min} = \min_{\tilde{p} \in \tilde{P}} \tilde{l}(\tilde{p})$, $\tilde{l}_{\max} = \max_{\tilde{p} \in \tilde{P}} \tilde{l}(\tilde{p})$, and $\varepsilon(\tilde{l}_{\max})$ is a regularization term defined as:

$$\varepsilon(\tilde{l}_{\max}) = \frac{\tilde{l}_{Avg}}{\tilde{l}_{\max} + \gamma}, \tag{17}$$

Table 1: Object-level AUROC and CV on Real3D-AD. Best and second-best are marked in **bold**.

| Method | Airplane | Car | Candybar | Chicken | Diamond | Duck | Fish | Gemstone | Seahorse | Shell | Starfish | Toffees | Avg. | CV |
|---|---|---|---|---|---|---|---|---|---|---|---|---|---|---|
| BTF(Raw) | 0.520 | 0.560 | 0.462 | 0.432 | 0.545 | 0.784 | 0.549 | 0.648 | 0.779 | 0.754 | 0.575 | 0.630 | 0.603 | 0.188 |
| BTF(FPFH) | 0.730 | 0.647 | 0.703 | 0.789 | 0.707 | 0.691 | 0.602 | 0.686 | 0.596 | 0.396 | 0.530 | 0.539 | 0.635 | 0.163 |
| M3DM(PointBert) | 0.407 | 0.506 | 0.442 | 0.673 | 0.627 | 0.466 | 0.556 | 0.617 | 0.494 | 0.577 | 0.528 | 0.562 | 0.538 | 0.141 |
| M3DM(PointMAE) | 0.434 | 0.541 | 0.450 | 0.683 | 0.602 | 0.433 | 0.540 | 0.644 | 0.495 | 0.694 | 0.551 | 0.552 | 0.552 | 0.157 |
| PatchCore(FPFH) | **0.882** | 0.590 | 0.565 | 0.837 | 0.574 | 0.546 | 0.675 | 0.370 | 0.505 | 0.589 | 0.441 | 0.541 | 0.593 | 0.237 |
| PatchCore(FPFH+Raw) | 0.848 | **0.777** | 0.626 | **0.853** | 0.784 | 0.628 | 0.837 | 0.359 | 0.767 | 0.663 | 0.471 | 0.570 | 0.682 | 0.222 |
| PatchCore(PointMAE) | 0.726 | 0.498 | 0.585 | 0.827 | 0.783 | 0.489 | 0.630 | 0.374 | 0.539 | 0.501 | 0.519 | 0.663 | 0.595 | 0.217 |
| CPMF | 0.632 | 0.518 | 0.718 | 0.640 | 0.640 | 0.554 | 0.840 | 0.349 | **0.843** | 0.392 | 0.582 | **0.845** | 0.625 | 0.256 |
| IMRNet | 0.762 | 0.711 | 0.755 | 0.780 | 0.905 | 0.517 | 0.880 | 0.674 | 0.604 | 0.665 | 0.674 | 0.774 | 0.725 | 0.144 |
| Reg3D-AD | 0.716 | 0.697 | 0.827 | **0.852** | 0.900 | 0.584 | 0.915 | 0.417 | 0.762 | 0.582 | 0.506 | 0.685 | 0.704 | 0.215 |
| R3D-AD | 0.772 | 0.696 | 0.713 | 0.714 | 0.685 | **0.909** | 0.692 | 0.665 | 0.720 | **0.840** | 0.701 | 0.700 | 0.734 | 0.094 |
| Group3AD | 0.744 | 0.728 | **0.847** | 0.786 | 0.932 | 0.679 | **0.976** | 0.539 | **0.841** | 0.585 | 0.562 | 0.796 | 0.751 | 0.179 |
| MC3D-AD | 0.850 | 0.749 | 0.830 | 0.715 | **0.955** | 0.831 | 0.865 | 0.560 | 0.716 | 0.803 | **0.766** | 0.738 | **0.782** | 0.121 |
| PO3AD | 0.804 | 0.654 | 0.785 | 0.686 | 0.801 | 0.820 | 0.859 | **0.693** | 0.756 | 0.800 | 0.758 | 0.771 | 0.766 | **0.076** |
| Ours | **0.867** | **0.773** | **0.850** | 0.821 | **0.962** | 0.847 | **0.924** | 0.719 | 0.767 | **0.799** | 0.800 | 0.831 | 0.830 | 0.078 |

Table 2: Point-level AUROC and CV on Real3D-AD. Best and second-best are marked in **bold**.

| Method | Airplane | Car | Candybar | Chicken | Diamond | Duck | Fish | Gemstone | Seahorse | Shell | Starfish | Toffees | Avg. | CV |
|---|---|---|---|---|---|---|---|---|---|---|---|---|---|---|
| BTF(Raw) | 0.520 | 0.560 | 0.462 | 0.432 | 0.545 | 0.784 | 0.549 | 0.648 | 0.779 | 0.754 | 0.575 | 0.630 | 0.603 | 0.188 |
| BTF(FPFH) | **0.738** | 0.708 | 0.864 | 0.693 | 0.882 | **0.875** | 0.709 | **0.891** | 0.512 | 0.571 | 0.501 | 0.815 | 0.730 | 0.187 |
| M3DM(PointBert) | 0.523 | 0.593 | 0.682 | **0.790** | 0.594 | 0.668 | 0.589 | 0.646 | 0.574 | 0.732 | 0.563 | 0.677 | 0.636 | **0.116** |
| M3DM(PointMAE) | 0.530 | 0.607 | 0.683 | 0.735 | 0.618 | 0.678 | 0.600 | 0.654 | 0.561 | 0.748 | 0.555 | 0.679 | 0.637 | **0.106** |
| PatchCore(FPFH) | 0.471 | 0.643 | 0.637 | 0.618 | 0.760 | 0.430 | 0.464 | **0.830** | 0.544 | 0.596 | 0.522 | 0.411 | 0.577 | 0.215 |
| PatchCore(FPFH+Raw) | 0.556 | 0.740 | 0.749 | 0.558 | 0.854 | 0.658 | 0.781 | 0.539 | 0.808 | 0.753 | 0.613 | 0.549 | 0.680 | 0.160 |
| PatchCore(PointMAE) | 0.579 | 0.610 | 0.635 | 0.683 | 0.776 | 0.439 | 0.714 | 0.514 | 0.660 | 0.725 | 0.641 | 0.727 | 0.642 | 0.144 |
| CPMF | 0.618 | **0.836** | 0.734 | 0.559 | 0.753 | 0.719 | **0.988** | 0.449 | **0.962** | 0.725 | **0.800** | 0.959 | 0.759 | 0.209 |
| Reg3D-AD | 0.716 | 0.697 | 0.827 | **0.852** | 0.900 | 0.584 | 0.915 | 0.417 | 0.762 | 0.582 | 0.506 | 0.685 | 0.704 | 0.215 |
| Group3AD | 0.636 | 0.610 | 0.738 | 0.759 | 0.862 | 0.631 | 0.836 | 0.564 | **0.827** | 0.625 | | 0.803 | 0.735 | 0.128 |
| MC3D-AD | 0.628 | 0.819 | **0.910** | 0.640 | 0.942 | 0.822 | 0.932 | 0.458 | 0.659 | 0.778 | 0.690 | **0.934** | 0.768 | 0.192 |
| Ours | **0.763** | 0.825 | 0.881 | 0.700 | 0.938 | 0.844 | 0.902 | 0.534 | 0.731 | **0.790** | 0.753 | 0.862 | **0.794** | 0.132 |

Here, $\gamma > 0$ denotes a set of hyperparameters. This formulation guarantees that when $\tilde{l}_{\max}$ is small (i.e., no point exhibits a strong deviation), the score range is compressed, thereby reducing the likelihood of false positives. In contrast, when $\tilde{l}_{\max}$ becomes sufficiently large, the score associated with the most deviated point asymptotically approaches 1, effectively highlighting high-confidence anomalies.

## 4 EXPERIMENTS

### 4.1 IMPLEMENTATION DETAILS.

The proposed EG3AD is first pretrained on ShapeNet55 (Chang et al., 2015) during the first stage, and then finetuned on PCAD datasets. We use the AdamW (Loshchilov & Hutter, 2017) optimizer for training, with an initial learning rate of 0.001 and a weight decay of 0.05. Each point cloud sample is first downsampled to 8,192 points to enable parallel training. It is then divided into 256 groups, with each group containing 32 points. These 256 patches are subsequently transformed into 256 tokens, each represented by a 384-dimensional embedding. The batch size is set to 80, and the maximum number of training epochs is 1,000. The model uses 3 stacked PCGC blocks and 3 stacked Transformer decoder layers. In CAS, the drop ratios for $P_{\text{low}}$, $P_{\text{mid}}$, and $P_{\text{high}}$ are empirically set to 4:3:3 on the Real3D-AD benchmark and 5:3:2 on the Anomaly-ShapeNet. All experiments are conducted using PyTorch 1.8.0 and CUDA 11.4 on 4 Tesla V100-SXM2-16GB GPUs.

### 4.2 MAIN RESULTS

EG3AD is compared against a comprehensive set of methods on the Real3D-AD and Anomaly-ShapeNet benchmarks, including both state-of-the-art ulti-class frameworks (e.g., MC3D-AD (Zhou et al., 2024), R3D-AD (Cheng et al., 2025a), IM-RNet (Li et al., 2024)) and recent single-class approaches (e.g., Reg3D-AD (Zhou et al., 2024), Group3AD (Zhu et al., 2024), PointCore (Zhao et al., 2024), PO3AD (Ye et al., 2024), CPMF (Cao et al., 2024)), along with several baseline methods reported in the Real3D-AD benchmark.

Table 3: Object-level AUROC, point-level AUROC, and CV on Anomaly ShapeNet, with the best and second-best results highlighted in **bold**.

| Method | Object-level | | Point-level | |
|---|---|---|---|---|
| | O-AUROC | O-CV | P-AUROC | P-CV |
| BTF(Raw) | 0.493 | 0.227 | 0.550 | 0.155 |
| BTF(FPFH) | 0.528 | 0.228 | 0.628 | 0.142 |
| M3DM | 0.616 | 0.218 | 0.616 | 0.130 |
| PatchCore(FPFH) | 0.568 | 0.188 | 0.580 | 0.202 |
| PatchCore(PointMAE) | 0.562 | 0.135 | 0.577 | 0.201 |
| CPMF | 0.559 | 0.187 | 0.573 | 0.145 |
| Reg3D-AD | 0.572 | 0.186 | 0.668 | 0.158 |
| MC3D-AD | 0.768 | 0.122 | / | / |
| R3D-AD | 0.749 | **0.078** | / | / |
| Group3AD | 0.577 | 0.154 | / | / |
| PO3AD | **0.839** | 0.113 | **0.899** | **0.082** |
| Ours | **0.856** | **0.096** | **0.878** | **0.062** |

Table 4: Ablation study on the effectiveness of each component in our framework. CAS, PCGC and FPM.

| CAS | PCGC | FPM | O-AUROC | P-AUROC |
|-----|------|-----|---------|---------|
| ✗ | ✗ | ✗ | 0.688 | 0.696 |
| ✓ | ✗ | ✗ | 0.702 | 0.719 |
| ✓ | ✓ | ✗ | 0.763 | 0.748 |
| ✓ | ✓ | ✓ | **0.830** | **0.794** |

Table 5: Ablation study on the complexity, computational cost, and performance of the PCGC encoder with varying numbers of PCGC blocks.

| Num of PCGC | Parameters | FLOPs | O-AUROC | P-AUROC |
|-------------|-----------|-------|---------|---------|
| 1 | 2.1M | 771M | 0.529 | 0.533 |
| 2 | 3.5M | 1142M | 0.702 | 0.688 |
| 3 | 4.8M | 1529M | 0.830 | 0.794 |
| 4 | 6.2M | 1892M | 0.837 | 0.797 |

Tables 1 and 2 present a detailed per-class comparison of the proposed EG3AD with state-of-the-art anomaly detection methods on the widely used Real3D-AD dataset. EG3AD achieves substantial improvements over prior approaches, securing the best or second-best results across multiple object categories. Ultimately, it attains an average object-level AUROC of 81.1% and an average point-level AUROC of 78.7%, outperforming all existing multi-class and single-class 3D anomaly detection methods.

Table 3 presents the performance of EG3AD compared to other methods on the Anomaly-ShapeNet benchmark. EG3AD exhibits strong results, achieving the highest average object-level AUROC of 85.6% and the second-highest average point-level AUROC of 87.8%.

In addition, EG3AD also obtains best or second best CV scores on both two benchmarks. These results indicate that EG3AD delivers consistent performance across diverse object categories, highlighting its strong generalization capability.

## 4.3 ABLATION STUDIES

### 4.3.1 EFFECTIVENESS OF MODULES

To validate the effectiveness of the proposed CAS, PCGC, and FPM modules, we build a vanilla model consisting of standard FPS and a standard Point Transformer encoder with six blocks, without FPM. Then, we progressively replace FPS with CAS, substitute the first three Point Transformer blocks with PCGC, and replace the last three blocks with the FPM module. The performance of each variant is reported in Table 4. The results demonstrate the effectiveness of all three components, with PCGC and FPM contributing especially significant improvements.

### 4.3.2 CHOICE OF CAS DROP RATE

After the initial sampling, CAS performs additional point dropping based on curvature, following specific drop ratios for low-, mid-, and high-curvature regions. As shown in Table 6, we first set up a balanced drop configuration as a baseline. Then, we progressively increase the drop rate for low-curvature points while keeping other factors fixed. The results reveal that moderately increasing the drop ratio for low-curvature regions helps improve detection performance. However, overly aggressive dropping leads to a sharp degradation in performance, indicating the importance of preserving sufficient representation of low-curvature regions. Empirically, each type of object has its own optimal drop rate, while in multi-class PCAD the drop rate is a compromise choice.

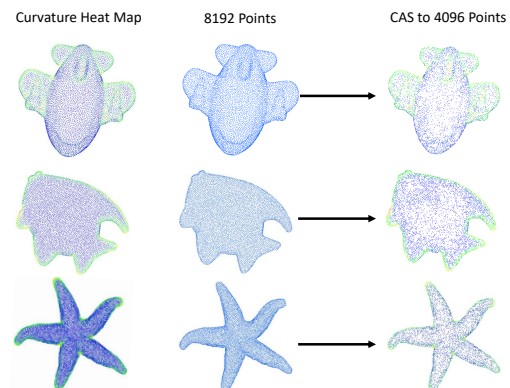

Figure 4: Visualization of effectiveness of CAP.

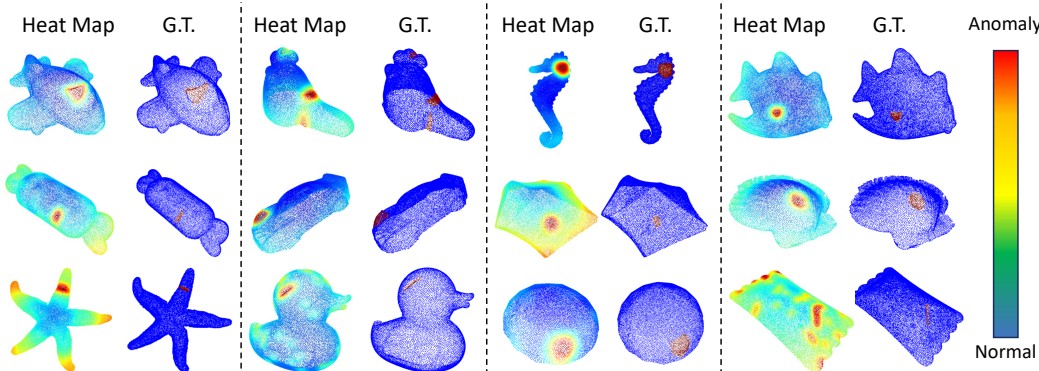

Figure 5: Point-level heatmap comparison between our proposed EG3AD and the Ground Truth (GT) on the Real3D-AD dataset. As indicated by the red regions in the heatmaps, EG3AD accurately detects and localizes anomalous areas within point clouds across diverse categories, showing its strong discriminative ability.

Table 6: Ablation study on the drop ratio settings of CAS. Moderate dropping of low-curvature points improves performance, while excessive dropping degrades results.

| Setting | Drop Ratio | | | Real3D-AD | | Anomaly-ShapeNet | |
|---|---|---|---|---|---|---|---|
| | $P_{low}$ | $P_{mid}$ | $P_{high}$ | O-AUROC | P-AUROC | O-AUROC | P-AUROC |
| 1 | 33.3% | 33.3% | 33.3% | 0.803 | 0.775 | 0.833 | 0.861 |
| 2 | 40.0% | 30.0% | 30.0% | 0.830 | 0.794 | 0.841 | 0.867 |
| 3 | 50.0% | 30.0% | 20.0% | 0.756 | 0.727 | 0.856 | 0.878 |
| 4 | 60.0% | 30.0% | 10.0% | 0.709 | 0.688 | 0.813 | 0.862 |
| 5 | 70.0% | 25.0% | 5.0% | 0.680 | 0.664 | 0.772 | 0.769 |

### 4.3.3 NUMBER OF PCGC BLOCKS

The number of PCGC blocks has a significant impact on model performance, capacity, and computational efficiency. To determine the optimal number of blocks, we conducted four experiments on the Real3D-AD benchmark. The results are presented in Table 5. We observe a substantial performance improvement when increasing the number of blocks from 2 to 3. Although using 4 blocks yields slightly better results than 3 blocks, the performance gain is marginal relative to the corresponding increase in model size and computational cost.

## 5 VISUALIZATION

Figure 5 shows the heatmap visualization results of EG3AD on Real3D-AD dataset. It can be observed that EG3AD accurately detects and localizes anomalous regions within the point cloud, clearly demonstrating its effectiveness. Additionally, Figure 4 demonstrates the sampling outcomes where the original point cloud was initially downsampled using FPS, followed by further downsampling through CAS. The visualization clearly shows that CAS preferentially preserves more sampling points in high-curvature regions (marked in red).

## 6 CONCLUSION

In this paper, we first analyze the limitations of directly using point transformers as encoders for reconstruction-based PCAD. We then propose EG3AD, which incorporates stronger prior biases and a more easily trainable encoder framework. Experimental results show that replacing the point transformer encoder with EG3AD yields both higher computational efficiency and stronger detection

performance, paving the way for advanced autonomous manufacturing applications. Future work can focus on optimizing the model for specific industrial scenarios.

## 7 USE OF LLMS

At the early stage of this work, we employed LLMs to check grammar and spelling and to perform automatic corrections. During the later stages of writing, we referred to LLMs for polishing certain sentences into more academic English. All outputs were subsequently reviewed and further refined by us. We clarify this here for transparency.

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

# APPENDIX

## CONTENTS OF APPENDIX

## A  MORE DATASET DETAILS

### A.1  REAL3D-AD DATASET

Real3D-AD is a high-resolution 3D point cloud dataset designed for real-world anomaly detection. It is captured using a PMAX-S130 optical system. Each object is scanned by rotating it 360° on a turntable, followed by a reverse-side scan. After manual alignment of front and back scans, an automatic stitching process is applied and repeated if necessary to generate complete point clouds.

Unlike RGB-D datasets, Real3D-AD provides only raw 3D point cloud data without color or depth maps. It contains 1,254 samples across 12 object categories: Airplane, Car, Candybar, Chicken, Diamond, Duck, Fish, Gemstone, Seahorse, Shell, Starfish, and Toffees. The dataset supports both training and evaluation for point cloud-based anomaly detection.

## A.2 ANOMALY-SHAPENET DATASET

Anomaly-ShapeNet is a synthetic dataset specifically designed for benchmarking 3D anomaly detection algorithms. It is constructed by injecting artificial anomalies into clean CAD models sourced from ShapeNet, enabling the creation of a large and diverse set of anomalous objects without requiring manual annotation or real-world data collection.

The dataset comprises a total of 1600 point cloud samples across 50 categories, including 40 standard categories and 10 auxiliary ones. For each object class, only four normal samples are provided for training, following the standard few-shot setting commonly adopted in 3DAD benchmarks. The test set for each class consists of 15 to 24 samples, including both normal and anomalous instances.

Each object is represented as a watertight mesh, which is then sampled into a point cloud containing between 8000 and 30000 points per sample. The anomalies are synthetically generated and include six types: bulge, concavity, crack, holes, cut, and broken, each designed to simulate different forms of realistic surface-level defects. The proportion of anomalous regions in a point cloud ranges from approximately 1% to 7% of the total points, ensuring that the anomalies are subtle and localized—posing a meaningful challenge for geometric perception and anomaly detection. This dataset provides a controlled yet diverse testing ground to evaluate both generalization and fine-grained detection capability of PCAD methods under limited supervision.

# B EVALUATION METRICS

## B.1 O-AUROC

The Receiver Operating Characteristic (ROC) curve is a graphical representation depicting the discriminative capability of a binary classifier as its decision threshold is adjusted. This curve is constructed by plotting the True Positive Rate (TPR), also known as sensitivity, against the False Positive Rate (FPR), which is equivalent to (1 - specificity), across a spectrum of threshold values. The Area Under the ROC Curve (AUROC) quantifies the overall performance of the classifier by measuring the area beneath the ROC curve, thus providing a consolidated metric across all potential classification thresholds. A model exhibiting perfect discrimination achieves an AUROC of 1.0, signifying its ability to flawlessly distinguish between the positive and negative classes across all possible thresholds.

The Overall AUROC (O-AUROC) specifically evaluates the classifier's global capacity to differentiate between normal and anomalous objects as complete instances. This metric proves particularly efficacious for assessing performance on datasets characterized by varying degrees of anomaly prevalence. By considering the entire object, O-AUROC ensures a robust evaluation of the classifier's practical utility in scenarios where anomalies may manifest in diverse ways and frequencies.

## B.2 P-AUROC

Analogous to the O-AUROC, the Point-level AUROC (P-AUROC) concentrates on evaluating the classifier's performance at a fine-grained level, specifically considering each individual point within a 3D point cloud. This metric is of paramount importance for applications necessitating detailed intra-object analysis, where the classification of each constituent point directly influences the overall efficacy of the detection system. P-AUROC furnishes a measure of the classifier's precision in classifying every point within an object, rendering it indispensable for tasks demanding high-resolution anomaly localization. Examples of such applications include manufacturing quality control or structural integrity assessments, where even minute anomalies can indicate substantial defects.

## B.3 CV

We introduce the *Coefficient of Variation (CV)* (Abdi, 2010) as an auxiliary metric to measure the performance consistency across multiple categories in multi-class PCAD tasks.

$$\mathrm{CV}_{\mathrm{AUROC}} = \frac{\sigma_{\mathrm{AUROC}}}{\mu_{\mathrm{AUROC}}}, \tag{18}$$

where $\sigma_{\text{AUROC}}$ and $\mu_{\text{AUROC}}$ denote the standard deviation and mean of AUROC scores across all object categories, respectively. A lower $\text{CV}_{\text{AUROC}}$ indicates more stable and balanced detection performance across categories.

## C FPM INFORMATION THEORY EXPLANATION

The FPM leverages the denoising effect of compression to enhance anomaly detection. Since 3D anomalies typically manifest as subtle geometric defects(see anomaly proportion), the compressed representation of an abnormal sample $\tilde{P}$ can be considered approximately equal to that of a normal sample $P$, i.e., $V_{\tilde{P}} \approx V_P$. This mechanism can be intuitively understood through the lens of mutual information, as detailed in the appendix.

This mechanism can be intuitively explained using mutual information. Assuming an abnormal sample can be modeled as the sum of a normal sample and a small perturbation:

$$\tilde{P} = P + \text{Noise} \tag{19}$$

$$I(P;\tilde{P},V_{\tilde{P}}) \approx I(P;\tilde{P},V_P) \tag{20}$$

$$I(P;\tilde{P},V_P) = I(P;\tilde{P}) + I(P;V_P \mid \tilde{P}) \tag{21}$$

$$I(P;\tilde{P},V_{\tilde{P}}) > I(P;\tilde{P}) \tag{22}$$

These expressions reveal that reconstruction under the guidance of FPM achieves higher mutual information with the normal pattern $P$ than reconstruction relying solely on the abnormal input $\tilde{P}$. The improvement is quantitatively characterized by the conditional mutual information term $I(P;V_P \mid \tilde{P})$.

## D MULTI STAGE TRAINING STRATEGY

As shown in Figure 6, we adopt a three-stage progressive training pipeline. In the first stage(Figure 6 a) the network is pretrained on a large-scale point cloud dataset using a variational autoencoder (VAE), enabling the model to learn generalizable geometric representations and acquire zero-shot reconstruction capability. In the second stage (Figure 6 b), Curvature-Aware Masked Autoencoder (CMAE) begins by masking geometrically simple (low-curvature) regions and progressively shifts toward more complex (high-curvature) ones as training advances.

Specifically, let the curvature vector of $M$ patches be denoted as $c = [c_1, c_2, \ldots, c_M]$, normalized to the range $[0,1]$. At each training epoch $t \in [1,T]$, we sample a threshold vector from a Gaussian distribution $\mathcal{N}(\mu_t, \sigma^2)$, where

$$\mu_t = \mu_{\min} + (\mu_{\max} - \mu_{\min}) \cdot \frac{t}{T}, \tag{23}$$

gradually increases from low to high values (e.g., $\mu_{\min} = 0.1$, $\mu_{\max} = 0.9$), and the variance $\sigma^2$ is fixed (e.g., $\sigma = 0.2$). This constructs a random masking mechanism based on a movable Gaussian function, where the Gaussian probability kernel progressively shifts toward high-curvature regions as the training epoch $t$ increases. Such a curriculum-style masking strategy not only facilitates more efficient model convergence, but also encourages comprehensive learning of reconstruction across varying curvature regions.

In the third stage (Figure 6 c), we adopt Curvature-aware Anomaly Autoencoder(CAAE), which further strengthens the model's discriminative capacity by automatically injecting synthetic anomalies. Similar to CMAE, CAAE follows the rule that transitions from flat to complex regions. The pseudo-anomaly generator randomly introduces small-scale dents, bumps, or Gaussian noise on the object surface, encouraging the model to reconstruct the corresponding normal patterns. This process effectively enhances the model's anomaly detection performance.

### D.1 DETAILS OF CAAE TRAINING

Table 7: Ablation study on synthetic anomaly generation strategies. Performance is reported in O-AUROC.

| Dataset | Gaussian | Bulge/Sink | Mixture |
|---|---|---|---|
| Real3D-AD | 0.791 | **0.830** | 0.813 |
| Anomaly-ShapeNet | 0.784 | **0.856** | 0.841 |

Table 8: Ablation study on synthetic anomaly generation strategies. Performance is reported in P-AUROC.

| Dataset | Gaussian | Bulge/Sink | Mixture |
|---|---|---|---|
| Real3D-AD | 0.761 | **0.794** | 0.774 |
| Anomaly-ShapeNet | 0.806 | **0.878** | 0.830 |

CAAE constitutes the third stage of the proposed multi-stage training strategy. It begins by injecting synthetic anomalies into geometrically simple (i.e., low-curvature) regions and progressively shifts toward more complex (i.e., high-curvature) regions as training progresses.

Specifically, let the curvature values of the $M$ patches be denoted as $c = [c_1, c_2, \ldots, c_M]$, corresponding to the patch set $\mathscr{C} = [\mathscr{C}_1, \mathscr{C}_2, \ldots, \mathscr{C}_M]$. The curvature values are normalized to the range $[0,1]$. At each training epoch $t \in [1, T]$, we sample a threshold vector from a Gaussian distribution $\mathscr{N}(\mu_t, \sigma^2)$, where

$$\mu_t = \mu_{\min} + (\mu_{\max} - \mu_{\min}) \cdot \frac{t}{T}, \qquad (24)$$

with $\mu_t$ increasing linearly over time (e.g., $\mu_{\min} = 0.1$, $\mu_{\max} = 0.9$), and the variance $\sigma^2$ fixed (e.g., $\sigma = 0.1$). The most probable patch sampled under the Gaussian distribution is used as the center for anomaly injection. The scale of each synthetic anomaly is set to 4% of the sample size, following the average anomaly proportion observed in the Real3D-AD and Anomaly-ShapeNet datasets.

We further investigate the impact of different synthetic anomaly generation strategies on the Real3D-AD and Anomaly-ShapeNet datasets. As shown in Table 7 and Table 8, the *Gaussian Defect* strategy injects only strong Gaussian noise; the *Pseudo Bulge/Sink* strategy generates only protrusions or indentations; and the *Mixture* strategy combines all three types, with 50% Gaussian noise, 25% bulges, and 25% indentations. The results show that the Bulge/Sink strategy achieves the best performance and is therefore adopted as the default anomaly generation method in our experiments.

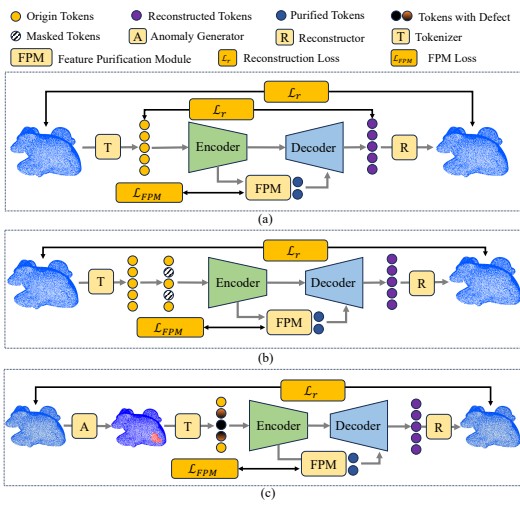

Figure 6: (a) Stage1: VAE Training. (b) Stage2: CMAE Training. (c) Stage3: CAAE Training.

## E  MORE IMPLEMENTATION INFORMATION

### E.1  TRAINING SETTING

The proposed EG3AD is first pretrained on ShapeNet55 during the first stage using VAE training. Then model is finetuned on PCAD datasets using CMAE training. Finally, We continue finetuning EG3AD on PCAD datasets using CAAE training. During these 3 stages above, we use the AdamW optimizer, with an initial learning rate of 0.001 and a weight decay of 0.05. We employ different data augmentation strategies across the three training stages. In the first stage, data augmentation includes random rotation, translation, scaling, and the addition of moderate Gaussian noise. In contrast, the latter two stages adopt a similar augmentation scheme but exclude Gaussian noise to preserve feature fidelity during fine-tuning. Each point cloud sample is first downsampled to 8,192 points to enable parallel training. It is then divided into 256 groups, with each group containing 32 points. The batch size is set to 80, and the maximum number of training epochs is 1,000.

## E.2 Loss Function Across Training Stages

In the first stage of training (VAE training) and the second stage of training (CMAE training), the total loss is defined as:

$$\mathcal{L}_{\text{total}} = \mathcal{L}_{\text{FPM}} + \mathcal{L}_{\text{r-patch}} + 100 \cdot \mathcal{L}_{\text{r-points}}. \tag{25}$$

In the third stage (CAAE training), since synthetic anomalies are directly introduced into the point cloud, the original tokens may contain corrupted information. To avoid misleading supervision, we discard $\mathcal{L}_{\text{FPM}}$ and $\mathcal{L}_{\text{r-patch}}$, and only retain the point-level reconstruction loss:

$$\mathcal{L}_{\text{total}} = 100 \cdot \mathcal{L}_{\text{points}}. \tag{26}$$

All experiments are conducted using PyTorch 1.8.0 and CUDA 11.4 on four Tesla V100-SXM2-16GB GPUs.

## E.3 Testing Setting

No data augmentation is applied during testing. Instead, all point clouds are uniformly normalized to fit within the range $[-1, 1]$ along all axes. Specifically, for a given point cloud $P = \{p_i\}_{i=1}^N$, each point $p = (x, y, z) \in P$ is scaled such that:

$$Max(x_{max} - x_{min},\ y_{max} - y_{min},\ z_{max} - z_{min}) = 1 \tag{27}$$

This ensures the largest spatial extent among the three axes is rescaled to 1, preserving the shape while standardizing the scale. For testing efficiency, each sample is downsampled to 8192 points before inference, and anomaly detection metrics are computed based on the downsampled samples. During sampling, the CAS module divides the sample into 256 patches with 32 points per patch, following the same procedure as in training.

## E.4 CAS Setting

Based on the sampled point cloud (containing 8192 points), CAS first uses FPS combined with KNN to uniformly divide the point cloud into 1024 patches, each containing 32 points. Curvature is then calculated for each patch. The patches are sorted by curvature and divided into three groups, from which a total of 768 patches are dropped. The drop ratios for $P_{\text{low}}$, $P_{\text{mid}}$, and $P_{\text{high}}$ are empirically set to 4:3:3 on the Real3D-AD benchmark and 5:3:2 on the Anomaly-ShapeNet benchmark.

## E.5 PCGC Setting

The point graph of PCGC is constructed based on the 256 patch centers grouped by CAS. For each token, 8 nearest patches are searched, i.e., the value of $k$ is set to 8. The fused token is then obtained through MLP and max pooling. The number of stacked PCGC Blocks is 3, which is consistent with decoder Blocks. The number of PCGC blocks is also discussed in the ablation study.

# F More Ablation Studies

## F.1 Stage Training Efectiveness.

To demonstrate the effectiveness of each phase in our progressive training pipeline, we record the model's performance after each stage. As is shown in Table 9, in Stage 1, the model is trained solely on a large-scale pretraining dataset in a zero-shot manner, resulting in limited anomaly detection ability. However, with the introduction of CMAE in Stage 2, the model's performance improves substantially. This improvement is further amplified in Stage 3 through the CAAE, showing the importance and complementary benefits of each training stage.

## F.2 Weight of Loss

Table 10 presents an ablation study on the weight assignment between the patch-level Chamfer Distance reconstruction loss and the point-level Chamfer Distance reconstruction loss. A higher

Table 9: Performance improvement across the three-stage progressive training pipeline.

| Training Stage | O-AUROC | P-AUROC |
|---|---|---|
| Stage 1 (VAE) | 0.552 | 0.643 |
| Stage 2 (CMAE) | 0.773 | 0.759 |
| Stage 3 (CAAE) | 0.830 | 0.794 |

Table 10: Ablation study of the loss weight for $L_{\text{r-points}}$ and $L_{\text{r-patches}}$ on Real3D-AD.

| Setting | $L_{\text{points}}$ | $L_{\text{patches}}$ | O-AUROC | P-AUROC |
|---|---|---|---|---|
| 1 | 1 | 1 | 0.821 | 0.784 |
| 2 | 1 | 100 | 0.816 | 0.779 |
| 3 | 100 | 1 | 0.830 | 0.794 |

weight is assigned to the point-level loss, as our reconstruction-based PCAD framework relies on point-wise reconstruction to compute point-level anomaly scores. This design choice is particularly critical in the third stage (CAAE), which heavily depends on accurate point-level reconstruction.

# G  PER-CLASS RESULTS ON ANOMALY-SHAPENET

Object level per-class results on Anomaly-Shapenet benchmark are shown in Table 11, Table 12 and Table 13. Point level per-class results on Anomaly-Shapenet benchmark are also shown in Table 14, Table 15 and Table 16.

Table 11: O-AUROC on Anomaly-ShapeNet (part I). Best and second-best are shown in **bold**.

| Method | ashtray0 | bag0 | bottle0 | bottle1 | bottle3 | bowl0 | bowl1 | bowl2 | bowl3 | bowl4 | bowl5 | bucket0 | bucket1 | cap0 |
|---|---|---|---|---|---|---|---|---|---|---|---|---|---|---|
| BTF (Raw) | 0.578 | 0.410 | 0.597 | 0.510 | 0.568 | 0.564 | 0.264 | 0.525 | 0.385 | 0.664 | 0.417 | 0.617 | 0.321 | 0.668 |
| BTF (FPFH) | 0.420 | 0.546 | 0.344 | 0.546 | 0.322 | 0.509 | 0.668 | 0.510 | 0.490 | 0.609 | 0.699 | 0.401 | 0.633 | 0.618 |
| M3DM | 0.577 | 0.537 | 0.574 | 0.637 | 0.541 | 0.634 | 0.663 | 0.684 | 0.617 | 0.464 | 0.409 | 0.309 | 0.501 | 0.557 |
| PatchCore (FPFH) | 0.587 | 0.571 | 0.604 | 0.667 | 0.572 | 0.504 | 0.639 | 0.615 | 0.537 | 0.494 | 0.558 | 0.469 | 0.551 | 0.580 |
| PatchCore | 0.591 | 0.601 | 0.513 | 0.601 | 0.650 | 0.523 | 0.629 | 0.458 | 0.579 | 0.501 | 0.593 | 0.593 | 0.561 | 0.589 |
| CPMF | 0.353 | 0.643 | 0.520 | 0.482 | 0.405 | **0.783** | 0.639 | 0.625 | 0.658 | 0.683 | 0.685 | 0.482 | 0.601 | 0.601 |
| Reg3D-AD | 0.597 | 0.706 | 0.486 | 0.695 | 0.525 | 0.671 | 0.525 | 0.490 | 0.348 | 0.663 | 0.593 | 0.610 | 0.752 | 0.693 |
| IMRNet | 0.671 | 0.660 | 0.552 | 0.700 | 0.640 | 0.681 | 0.702 | 0.685 | 0.599 | 0.676 | **0.710** | 0.580 | **0.771** | 0.737 |
| R3D-AD | 0.833 | 0.720 | 0.733 | 0.737 | 0.781 | 0.819 | 0.778 | 0.741 | 0.767 | 0.744 | 0.656 | 0.683 | 0.756 | 0.822 |
| PO3AD | **1.000** | **0.833** | **0.900** | **0.933** | **0.926** | 0.922 | 0.829 | **0.833** | 0.881 | **0.981** | **0.849** | 0.853 | 0.787 | **0.877** |
| MC3D-AD | 0.962 | 0.805 | 0.795 | 0.709 | 0.756 | **0.930** | **0.978** | 0.719 | **0.885** | 0.911 | 0.754 | **0.898** | 0.784 | 0.793 |
| Ours | **0.989** | 0.843 | 0.812 | 0.747 | 0.902 | **0.950** | **0.989** | 0.833 | **0.917** | **0.962** | 0.792 | **0.876** | 0.699 | **0.826** |

Table 12: O-AUROC on Anomaly-ShapeNet (part II). Best and second-best are shown in **bold**.

| Method | cap3 | cap4 | cap5 | cup0 | cup1 | eraser0 | headset0 | headset1 | helmet0 | helmet1 | helmet2 | helmet3 | jar0 | micro. |
|---|---|---|---|---|---|---|---|---|---|---|---|---|---|---|
| BTF (Raw) | 0.527 | 0.468 | 0.373 | 0.403 | 0.521 | 0.525 | 0.378 | 0.515 | 0.553 | 0.349 | 0.602 | 0.526 | 0.420 | 0.563 |
| BTF (FPFH) | 0.522 | 0.520 | 0.586 | 0.586 | 0.610 | 0.719 | 0.520 | 0.490 | 0.571 | 0.719 | 0.542 | 0.444 | 0.424 | 0.671 |
| M3DM | 0.423 | 0.777 | 0.639 | 0.539 | 0.556 | 0.627 | 0.577 | 0.617 | 0.526 | 0.427 | 0.623 | 0.374 | 0.441 | 0.357 |
| PatchCore (FPFH) | 0.453 | 0.757 | **0.790** | 0.600 | 0.586 | 0.657 | 0.583 | 0.637 | 0.546 | 0.484 | 0.425 | 0.404 | 0.472 | 0.388 |
| PatchCore | 0.476 | 0.727 | 0.538 | 0.610 | 0.556 | 0.677 | 0.591 | 0.627 | 0.556 | 0.552 | 0.447 | 0.424 | 0.483 | 0.488 |
| CPMF | 0.551 | 0.553 | 0.697 | 0.497 | 0.499 | 0.689 | 0.643 | 0.458 | 0.555 | 0.589 | 0.462 | 0.520 | 0.610 | 0.509 |
| Reg3D-AD | 0.725 | 0.643 | 0.467 | 0.510 | 0.538 | 0.343 | 0.537 | 0.610 | 0.600 | 0.381 | 0.614 | 0.367 | 0.592 | 0.414 |
| IMRNet | 0.775 | 0.652 | 0.652 | 0.643 | 0.757 | 0.548 | 0.720 | 0.676 | 0.597 | 0.600 | 0.641 | 0.573 | 0.780 | 0.755 |
| R3D-AD | 0.730 | 0.681 | 0.670 | 0.776 | 0.757 | 0.890 | 0.738 | 0.795 | 0.757 | 0.720 | 0.633 | 0.707 | 0.838 | 0.762 |
| PO3AD | **0.859** | 0.792 | 0.670 | **0.871** | 0.833 | **0.995** | 0.808 | **0.923** | **0.762** | **0.961** | **0.869** | 0.754 | 0.866 | 0.776 |
| MC3D-AD | 0.701 | **0.835** | **0.761** | 0.743 | **0.952** | 0.776 | **0.862** | **0.886** | 0.672 | **1.000** | 0.609 | **0.979** | **0.971** | **0.919** |
| Ours | **0.872** | 0.763 | 0.760 | **0.894** | **0.931** | **0.915** | **0.813** | 0.881 | **0.776** | 0.862 | **0.893** | **0.943** | **0.914** | 0.842 |

# H  MORE VISUALIZATIONS RESULTS

Figure 7 shows the heatmap visualization results of EG3AD on Anomaly-Shapenet dataset. It can be observed that EG3AD accurately detects and localizes anomalous regions within the point cloud, clearly demonstrating its effectiveness.

Table 13: O-AUROC on Anomaly-ShapeNet (part III). Best and second-best are shown in **bold**.

| Method | shelf0 | tap0 | tap1 | vase0 | vase1 | vase2 | vase3 | vase4 | vase5 | vase7 | vase8 | vase9 | Avg. | CV |
|---|---|---|---|---|---|---|---|---|---|---|---|---|---|---|
| BTF (Raw) | 0.164 | 0.525 | 0.573 | 0.531 | 0.549 | 0.410 | 0.717 | 0.425 | 0.585 | 0.448 | 0.424 | 0.564 | 0.493 | 0.227 |
| BTF (FPFH) | 0.609 | 0.560 | 0.546 | 0.342 | 0.219 | 0.546 | 0.699 | 0.510 | 0.409 | 0.518 | 0.668 | 0.268 | 0.528 | 0.228 |
| M3DM | 0.564 | 0.754 | 0.739 | 0.423 | 0.427 | 0.737 | 0.439 | 0.476 | 0.317 | 0.657 | 0.663 | 0.663 | 0.552 | 0.218 |
| PatchCore (FPFH) | 0.494 | 0.753 | 0.766 | 0.455 | 0.423 | 0.721 | 0.449 | 0.506 | 0.417 | 0.693 | 0.662 | 0.660 | 0.568 | 0.188 |
| PatchCore | 0.523 | 0.458 | 0.538 | 0.447 | 0.552 | 0.741 | 0.460 | 0.516 | 0.579 | 0.650 | 0.663 | 0.629 | 0.562 | 0.135 |
| CPMF | 0.685 | 0.359 | 0.697 | 0.451 | 0.345 | 0.582 | 0.582 | 0.514 | 0.618 | 0.397 | 0.529 | 0.609 | 0.559 | 0.187 |
| Reg3D-AD | 0.688 | 0.676 | 0.641 | 0.533 | 0.702 | 0.605 | 0.650 | 0.500 | 0.520 | 0.462 | 0.620 | 0.594 | 0.572 | 0.186 |
| IMRNet | 0.603 | 0.676 | 0.696 | 0.533 | 0.757 | 0.614 | 0.700 | 0.524 | 0.676 | 0.635 | 0.630 | 0.594 | 0.659 | 0.103 |
| R3D-AD | 0.696 | 0.736 | 0.788 | 0.729 | 0.752 | 0.742 | 0.630 | 0.757 | 0.771 | 0.721 | 0.718 |  | 0.749 | 0.078 |
| PO3AD | 0.573 | 0.745 | 0.681 | **0.858** | 0.742 | **0.952** | **0.821** | 0.675 | 0.852 | **0.966** | **0.739** | **0.830** | 0.839 | 0.113 |
| MC3D-AD | **0.841** | **0.945** | **0.970** | 0.821 | **0.857** | 0.929 | 0.761 | **0.876** | **0.976** | **0.938** | 0.670 | 0.736 | **0.842** | 0.122 |
| Ours | **0.703** | 0.765 | **0.927** | **0.861** | **0.973** | **0.942** | **0.856** | 0.779 | **0.934** | 0.818 | 0.653 | **0.817** | **0.856** | **0.096** |

Table 14: P-AUROC on Anomaly-ShapeNet (part I). Best and second-best are shown in **bold**.

| Method | ashtray0 | bag0 | bottle0 | bottle1 | bottle3 | bowl0 | bowl1 | bowl2 | bowl3 | bowl4 | bowl5 | bucket0 | bucket1 | cap0 |
|---|---|---|---|---|---|---|---|---|---|---|---|---|---|---|
| BTF (Raw) (CVPR 23') | 0.512 | 0.430 | 0.551 | 0.491 | 0.720 | 0.524 | 0.464 | 0.426 | 0.685 | 0.563 | 0.517 | 0.617 | 0.686 | 0.524 |
| BTF (FPFH) | 0.624 | 0.746 | 0.641 | 0.549 | 0.622 | 0.710 | 0.768 | 0.518 | 0.590 | 0.679 | 0.699 | 0.401 | 0.633 | 0.730 |
| M3DM (CVPR 23') | 0.577 | 0.637 | 0.663 | 0.637 | 0.532 | 0.658 | 0.663 | 0.694 | 0.657 | 0.624 | 0.489 | 0.698 | 0.699 | 0.531 |
| PatchCore (FPFH) (CVPR 22') | 0.597 | 0.574 | 0.654 | 0.687 | 0.512 | 0.524 | 0.531 | 0.625 | 0.327 | 0.720 | 0.358 | 0.459 | 0.571 | 0.472 |
| PatchCore (PointMAE) | 0.495 | 0.674 | 0.553 | 0.606 | 0.653 | 0.527 | 0.524 | 0.515 | 0.581 | 0.501 | 0.562 | 0.586 | 0.574 | 0.544 |
| CPMF (PR 24') | 0.615 | 0.655 | 0.521 | 0.571 | 0.435 | 0.745 | 0.488 | 0.635 | 0.641 | 0.683 | 0.684 | 0.486 | 0.601 | 0.601 |
| Reg3D-AD (NeurIPS 23') | 0.698 | 0.715 | 0.886 | 0.696 | 0.525 | 0.775 | 0.615 | 0.593 | 0.654 | 0.800 | 0.691 | 0.619 | 0.752 | 0.632 |
| IMRNet (CVPR 24') | 0.671 | 0.668 | 0.556 | 0.702 | 0.641 | 0.781 | 0.705 | 0.684 | 0.599 | 0.576 | 0.715 | 0.585 | 0.774 | 0.715 |
| PO3AD | **0.962** | **0.949** | **0.912** | 0.844 | **0.880** | **0.978** | **0.914** | **0.918** | **0.935** | **0.967** | **0.941** | 0.755 | **0.899** | **0.957** |
| Ours | **0.952** | **0.914** | **0.923** | **0.876** | **0.893** | **0.914** | **0.872** | **0.898** | **0.857** | **0.953** | **0.914** | **0.845** | **0.927** | **0.931** |

Table 15: P-AUROC on Anomaly-ShapeNet (part II). Best and second-best are shown in **bold**.

| Method | cap3 | cap4 | cap5 | cup0 | cup1 | eraser0 | headset0 | headset1 | helmet0 | helmet1 | helmet2 | helmet3 | jar0 | micro. |
|---|---|---|---|---|---|---|---|---|---|---|---|---|---|---|
| BTF (Raw) | 0.687 | 0.469 | 0.373 | 0.632 | 0.561 | 0.637 | 0.578 | 0.475 | 0.504 | 0.449 | 0.605 | 0.700 | 0.423 | 0.583 |
| BTF (FPFH) | 0.658 | 0.524 | 0.586 | 0.790 | 0.619 | 0.719 | 0.620 | 0.591 | 0.575 | 0.749 | 0.643 | 0.724 | 0.427 | 0.675 |
| M3DM | 0.605 | 0.718 | 0.655 | 0.715 | 0.556 | 0.710 | 0.581 | 0.585 | 0.599 | 0.427 | 0.623 | 0.655 | 0.541 | 0.358 |
| PatchCore (FPFH) | 0.653 | 0.595 | **0.795** | 0.655 | 0.596 | 0.810 | 0.583 | 0.464 | 0.548 | 0.489 | 0.455 | 0.737 | 0.478 | 0.488 |
| PatchCore | 0.488 | 0.725 | 0.545 | 0.510 | **0.856** | 0.378 | 0.575 | 0.423 | 0.580 | 0.562 | 0.651 | 0.615 | 0.487 | **0.886** |
| CPMF | 0.551 | 0.553 | 0.551 | 0.497 | 0.509 | 0.689 | 0.699 | 0.458 | 0.555 | 0.542 | 0.515 | 0.520 | 0.611 | 0.545 |
| Reg3D-AD | 0.718 | 0.815 | 0.467 | 0.685 | 0.698 | 0.755 | 0.580 | 0.626 | 0.600 | 0.624 | 0.825 | 0.620 | 0.599 | 0.599 |
| IMRNet | 0.706 | 0.753 | 0.742 | 0.643 | 0.688 | 0.548 | 0.705 | 0.476 | 0.598 | 0.604 | 0.644 | 0.663 | 0.765 | 0.742 |
| PO3AD | **0.948** | **0.940** | 0.864 | **0.909** | **0.932** | **0.974** | 0.823 | 0.907 | **0.878** | **0.948** | **0.932** | 0.846 | 0.871 | 0.810 |
| Ours | 0.925 | 0.912 | **0.873** | 0.877 | 0.911 | 0.914 | **0.936** | **0.915** | 0.845 | 0.896 | 0.904 | **0.873** | **0.896** | 0.792 |

Table 16: P-AUROC on Anomaly-ShapeNet (part III). Best and second-best are shown in **bold**.

| Method | shelf0 | tap0 | tap1 | vase0 | vase1 | vase2 | vase3 | vase4 | vase5 | vase7 | vase8 | vase9 | Average | CV |
|---|---|---|---|---|---|---|---|---|---|---|---|---|---|---|
| BTF (Raw) | 0.464 | 0.527 | 0.564 | 0.618 | 0.549 | 0.403 | 0.602 | 0.613 | 0.585 | 0.578 | 0.550 | 0.564 | 0.550 | 0.155 |
| BTF (FPFH) | 0.619 | 0.568 | 0.596 | 0.642 | 0.619 | 0.646 | 0.699 | 0.710 | 0.429 | 0.540 | 0.662 | 0.568 | 0.628 | 0.142 |
| M3DM | 0.554 | 0.654 | 0.712 | 0.608 | 0.602 | 0.737 | 0.658 | 0.655 | 0.642 | 0.517 | 0.551 | 0.663 | 0.616 | 0.130 |
| PatchCore (FPFH) | 0.613 | 0.733 | **0.768** | 0.655 | 0.453 | 0.721 | 0.430 | 0.505 | 0.447 | 0.693 | 0.575 | 0.663 | 0.580 | 0.202 |
| PatchCore | 0.543 | **0.858** | 0.541 | 0.677 | 0.551 | 0.742 | 0.465 | 0.523 | 0.572 | 0.651 | 0.364 | 0.423 | 0.577 | 0.201 |
| CPMF | 0.783 | 0.458 | 0.657 | 0.458 | 0.486 | 0.582 | 0.582 | 0.514 | 0.651 | 0.504 | 0.529 | 0.545 | 0.573 | 0.145 |
| Reg3D-AD | 0.688 | 0.589 | **0.741** | 0.548 | 0.602 | 0.405 | 0.511 | 0.755 | 0.624 | 0.881 | 0.811 | 0.694 | 0.668 | 0.158 |
| IMRNet | 0.605 | 0.681 | 0.699 | 0.535 | 0.685 | 0.614 | 0.401 | 0.524 | 0.682 | 0.593 | 0.635 | 0.691 | 0.650 | 0.126 |
| PO3AD | 0.663 | 0.783 | 0.692 | **0.955** | **0.882** | **0.978** | 0.884 | 0.902 | **0.937** | **0.982** | **0.950** | **0.952** | **0.899** | 0.082 |
| Ours | **0.831** | 0.847 | **0.717** | 0.875 | 0.861 | 0.762 | **0.906** | **0.917** | 0.800 | 0.926 | 0.784 | 0.763 | 0.878 | **0.062** |

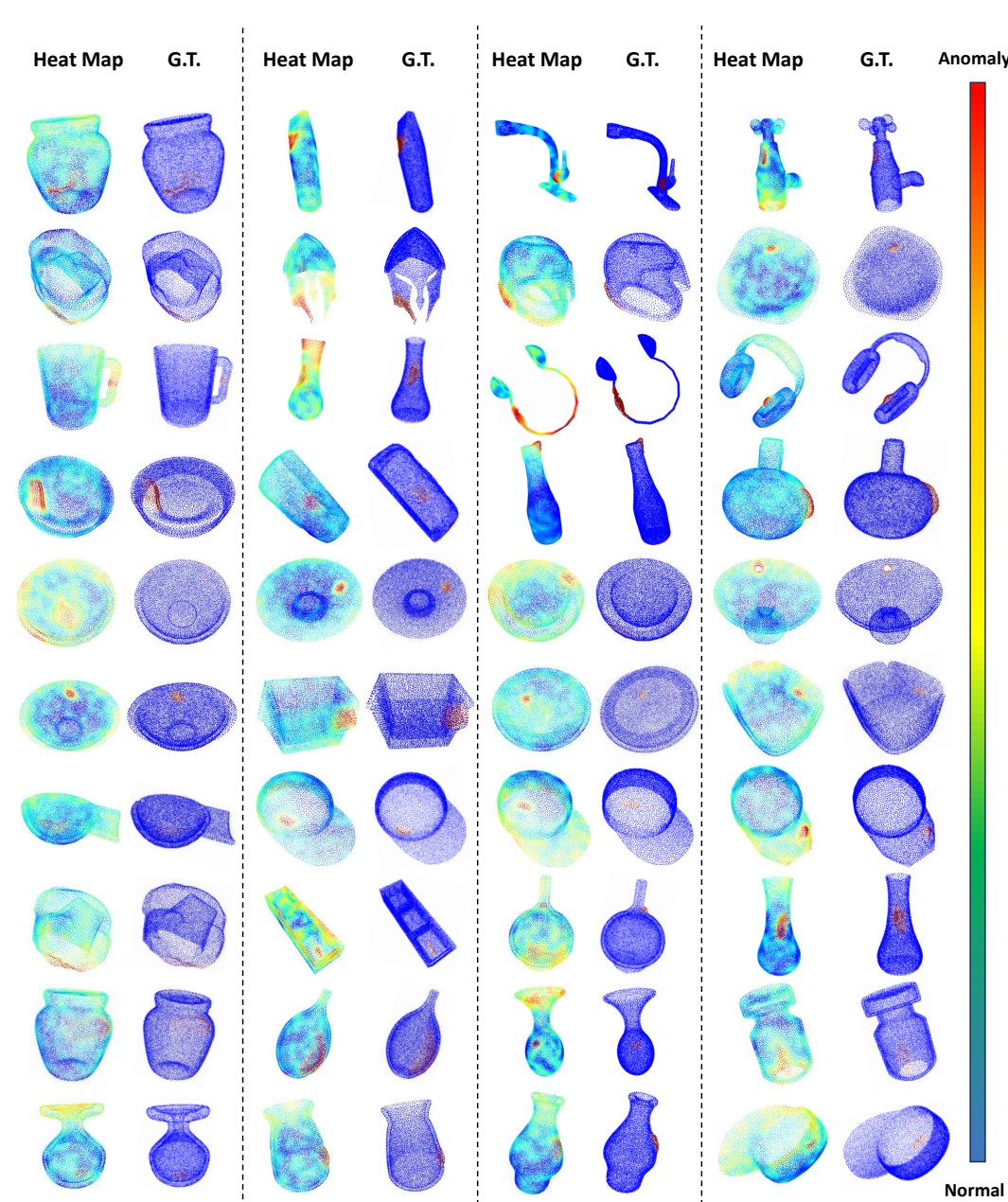

Figure 7: Point-level heatmap comparison between our proposed EG3AD and the Ground Truth (GT) on the Anomaly-Shapenet dataset. As indicated by the red regions in the heatmaps, EG3AD accurately detects and localizes anomalous areas within point clouds across diverse categories, showing its strong discriminative ability.

