# OpenReview forum: "EG3AD: An Efficient Geometry-Aware Encoding Framework for Reconstruction-Based Multi-Class Point Cloud Anomaly Detection"
_ICLR.cc/2026/Conference — Submitted to ICLR 2026_

### Official Review · Reviewer_22cG · 2025-10-15

**Soundness:** 3
**Presentation:** 2
**Contribution:** 2
**Rating:** 4
**Confidence:** 4

**Summary:**

This paper focuses on multi-class point cloud anomaly detection. The motivation is to address the large computational cost of the Transformer architecture and to keep fine-grained details in the point cloud sampling step. To this end, it proposes a strategy to sample low, mid, and high curvature points at different ratios and proposes a KNN-enhanced network to reduce the computational cost. The proposed method achieves leading performance on two major 3D anomaly detection benchmarks, including Real3D-AD and Anomaly-ShapeNet.

**Strengths:**

1) The motivation of this paper is clear. It is reasonable to address the computational cost for 3D anomaly detection, as the number of 3D points can be very large for high-resolution point clouds of a large object.

2) The proposed method demonstrates SOTA performance on leading benchmarks for 3D anomaly detection.

**Weaknesses:**

1) While the paper aims for 'efficient' 3D anomaly detection, its claims regarding memory and computational cost are not fully substantiated due to a lack of comparison with previous methods. The analysis of computational cost is incomplete; although a figure shows that the KNN-based PCGC module has an O(N) cost, this analysis omits the decoder and anomaly calculation modules, which add overhead during inference. To validate the claim of efficiency, it is crucial to provide a holistic comparison of the entire model's computational, memory, and time costs against other state-of-the-art methods.

2) The proposed curvature-based sampling strategy is a heuristic method reliant on pre-defined hyperparameters. According to the ablation study, performance is highly sensitive to these hyperparameters, which must be tuned for different datasets. This raises concerns about the module's effectiveness and generalization capability. The dependence on 'cherry-picked' hyperparameters may lead to overfitting on benchmarks and limit the method's applicability to unseen, real-world scenarios.

3) The paper contains several typos, errors, and unclear statements that need to be addressed:

- Figure 2a: Appears to be incomplete, with data points clipped from view.

- Line 93: The term 'cross-class metric' is undefined. If this means training on one class and inferring on others, this experimental setup is not presented in the paper. Please clarify.

- Lines 157-158: The definitions of m and m' seem inconsistent with their usage in Figure 3. Please check and correct these notations.

- Line 157: The citation for PointNet appears to be incorrect.

- Equation 14: The notation e is used without being defined.

- Line 320: The hyperparameter γ (gamma) is not defined, and its sensitivity on performance is not discussed.

- Tables: The formatting convention of bolding both the best and second-best results simultaneously is confusing. A clearer distinction is recommended.

4) According to the implementation details, the model is pre-trained on ShapeNet55. Since the Anomaly-ShapeNet test set contains objects from the ShapeNet dataset, this raises a concern about potential data leakage. Please clarify whether any overlap exists and how this issue was handled.

5) Regarding the ablation study on the drop ratio settings for CAS, the baseline performance is missing. What is the model's performance when no points are dropped (i.e., a drop ratio of 0)? This baseline is necessary to evaluate whether the sampling strategy provides a definitive benefit.

**Questions:**

Please see the questions in the weaknesses section.

**Details Of Ethics Concerns:**

Some figures in the appendix (e.g., Figures 6 and 7) appear to be taken directly from the original papers for the Real3D-AD and Anomaly-ShapeNet datasets. This is a severe violation of academic publishing standards.

---

> ### Author Response · Authors · 2025-11-27
> **Reply to Reviewer 22cG**
>
> Thanks for your evaluation of our work! Below is our response to the seven concerns you raised in questions:
>
> ## Response to Ethics Concern:
> We have received the letter from the Area Chair (AC) and have explained the issue. We have removed the figures and tables related to the ethics concern mentioned.
>
> ## Response to the 1st Concern:
> Following your advice, we have tested the overall efficiency of EG3AD (which includes CAS, PCGC, and FPM in the encoder, as well as the decoder) in terms of Parameters, Inference Time, and FLOPs. This serves as a supplement to the ablation study on the complexity, computational cost, and performance of the PCGC encoder with varying numbers of PCGC blocks, which only considered the PCGC. The experimental setup is consistent with the tests on the Real3D-AD dataset. The experimental results show that the encoder proposed by EG3AD enhances the encoding capability and reduces the dependence on Transformer blocks, leading to a significant improvement in parameter efficiency. However, the introduction of CAS requires solving PCA for each patch, which increases inference time. In terms of FLOPs, EG3AD shows some improvement compared to point transformer-based methods.
>
> Detailed results are provided below:
>
> ### Table: Efficiency Test of Parameters, Inference Time, and FLOPs
>
> | **Setting**      | **FLOPs**  | **Throughput (samples/s)** | **Parameters (M)** |
> |------------------|------------|----------------------------|--------------------|
> | **Setting 1**    | 9.531G     | 65.88                      | 22.1M              |
> | **Setting 2**    | 9.531G     | 48.47                      | 22.1M              |
> | **Setting 3**    | 8.72G      | 47.4                       | 7.793M             |
>
>
> ## Response to the 2nd Concern:
> Typical curvature-aware downsampling(like proposed in Towards Scalable 3D Anomaly Detection and Localization: A Benchmark via 3D Anomaly Synthesis and A Self-Supervised Learning Network) involves patch division, curvature calculation, and the selection of top-k points. In our approach, we made slight modifications to the selection of top-k points. Specifically, we categorize points into three tiers based on their curvature ranking and discard points from each tier proportionally. This approach helps to avoid the risk of losing too much information from low-curvature regions, where anomaly can likely to appear.
>
> In fact, many works follow the "one class, one model" paradigm, meaning that a separate model needs to be trained for each object type. Our method aims to address the generalization issue inherent in this approach. Experimental results have shown that we have succeeded in reaching this goal, and in some cases, our method even outperforms certain "one class, one model" methods in terms of testing metrics for some categories. Although we fine-tuned certain parameters for optimal performance on the test dataset, even without this fine-tuning, our method already demonstrates significant improvements in both prediction accuracy and generalization compared to "one class, one model" approaches.
>
>
>
> ## Response to the 3rd Concern:
>
> 1. The data in Figure 2a is not clipped from the view; the dashed area simply reflects the trend.
> 2. The cross-class balance metric refers to the Coefficient of Variation (CV), which is introduced in Appendix B.3. It measures the balance of the model’s predictions across different classes. A high CV indicates that there are large discrepancies in the predictions for different classes.
> 3. In Lines 157-158, the definitions of \(m\) and \(m'\) are consistent and correct. CAS first performs downsampling to obtain \(m'\) central points, and from these \(m'\) central points, \(m'\) patches are formed. After computing the curvature for each patch, a certain number of patches are dropped proportionally from the high, medium, and low curvature regions, resulting in \(m\) patches. These patches are then encoded to produce \(m\) embeddings, which are input into PCGC.
> 4. Thank you for pointing that out. We have corrected the citation format for PointNet.
> 5. Thank you. We have added a description for \(\mathbf{e}\): "Let \(\mathbf{e}_{Q_i}^{(0)}\) and \(\mathbf{e}_{D_j}^{(L)}\) represent the \(i\)-th token of the input to the 1st layer's Encoder and the \(j\)-th token of the output of the last layer's Decoder, respectively."
> 6. Actually, \(\gamma\) is a hyperparameter that varies significantly depending on factors such as point cloud scale, the number of points in the point cloud, and anomaly scale. Therefore, we set it to 0.05 based on experimental results.
> 7. The reason we bolded both the best and second-best values is because our method exhibits a phenomenon where, although it does not achieve the best result in multiple classes, it attains the highest average value. This is due to the balance of our method across different classes (as reflected in the better CV value). Therefore, simply bolding the highest value would not adequately reflect this aspect.

---

> > ### Author Response · Authors · 2025-11-27
> > **Reply to Reviewer 22cG(Continue)**
> >
> > ## Response to the 4th Concern:
> > We agree that there is a potential risk of data leakage. To investigate this issue, we designed the following experiment: we included the entire training set of Anomaly-Shapenet and the Real3D-AD dataset in the pretraining process. Afterward, we fine-tuned the model and evaluated its performance on these two datasets.
> >
> > For Anomaly-Shapenet, the prediction results remained unchanged (point-level AUROC = 0.878), which initially raised our suspicion. Surprisingly, the test performance on Real3D-AD also showed no significant change: the point-level AUROC only increased from 0.794 to 0.796, which is far below our expectation. We believe this is because, during pretraining, the model primarily learns generalizable representations of point clouds, rather than task-specific knowledge. The ability to reconstruct the normal form from anomalous samples is primarily acquired during the fine-tuning stage, and this reconstruction ability is precisely what is most crucial for anomaly detection.
> >
> > Actually, we follow the same experimental setting as the official benchmark for the Anomaly-Shapenet dataset. For example, **IMRNet**, which is introduced in the same paper as Anomaly-Shapenet, is pretrained on ShapeNet55 (as indicated in their implementation details). Other classical methods in the Anomaly-Shapenet benchmark, such as **Reg3DAD**, **M3DM (PointMAE)**, **M3DM (PointBert)**, and **Patchcore (PointMAE)**, also rely on pretrained checkpoints provided by PointMAE or PointBert on GitHub. These checkpoints are first pretrained on ShapeNet and then fine-tuned on other downstream task datasets.
> >
> > ## Response to the 5th Concern:
> > We indeed conducted the FPS experiment. Table 6 in the main text: "Ablation study on the drop ratio settings of CAS. Moderate dropping of low-curvature points improves performance, while excessive dropping degrades results." Setting 1 is actually the one you need for a pure FPS case, as Setting A has the same drop rate across the low, medium, and high curvature regions.
> >
> > If you have any further questions or clarifications, please feel free to let me know.

---

> ### Author Response · Authors · 2025-12-01
> **Summary to My Rebuttal**
>
> This paper proposes an encoder specifically designed for multi-class anomaly detection tasks, focusing on improving three encoding stages: tokenizing, shallow feature encoding, and deep feature encoding. To address these stages, we introduce CAS, PCGC, and FPM. With the combination of these components and the use of only three transformer decoder blocks, EG3AD achieves excellent detection performance.
>
> The reviewers agree the SOTA performance of our method and raised concerns regarding inference time, parameter utilization, and computational complexity. We have supplemented additional experiments to address these concerns.
>
> For the minor questions and suggestions raised by the reviewer, we have responded accordingly or further clarified them in the revised version. The reviewer also raised the same concern as in reviewer whZ9 regarding potential data leakage from using ShapeNet. We have addressed this concern with experimental evidence. Additionally, the ablation studies for CAS and FPM, which is required by reviewer, were already included in the original manuscript, so we just pointed this out to the reviewer.

---

### Official Review · Reviewer_2PbA · 2025-10-30

**Soundness:** 3
**Presentation:** 3
**Contribution:** 3
**Rating:** 4
**Confidence:** 4

**Summary:**

The paper identifies the high complexity of existing methods and proposes a multi-class point cloud anomaly detection method. The method consists of three modules: CAS, PCGC, and FPM. Experiments demonstrate the effectiveness of the proposed modules.

**Strengths:**

The paper is well-organized, and each component of the method is clearly described and easy to understand. The superiority of the proposed method is validated on public datasets, including Real3D and Anomaly-ShapeNet. Ablation studies further confirm its effectiveness.

**Weaknesses:**

The related work section is not sufficiently comprehensive. For instance, “Boosting global-local feature matching via anomaly synthesis for multi-class point cloud anomaly detection” also addresses a similar multi-class anomaly detection setting and should be discussed for completeness.

The novelty of the proposed method remains questionable. The CAS module appears highly similar to curvature-based downsampling, and it is unclear what the essential differences are. Likewise, the distinction between PCGC and PointNet++ is not clearly justified.

Although the paper claims to reduce the high computational cost of Point Transformer, the proposed architecture still employs multiple Transformer decoders, and the PCGC module contains numerous fully connected layers. These components introduce considerable computational overhead, and it is doubtful whether the method is indeed more efficient in practice.

The visualization in Fig. 4 also seems problematic. The curvature maps show a smooth radial transition from the center outward, which is unlikely to be realistic. For example, the regions between the arms of a starfish should exhibit noticeable curvature changes, yet they are visualized as uniformly blue.

In addition, the relationship between curvature estimation and anomaly detection performance is not clearly analyzed or discussed.

It is also recommended to provide more qualitative comparisons with other methods to more convincingly demonstrate the superiority of the proposed approach.

Finally, since the method is evaluated under a multi-class setting, it is unclear how the results for other methods were obtained. It would be more convincing if the authors also reported single-class performance and compared against the latest state-of-the-art baselines, for which more reported results are available.

**Questions:**

Please refer to the issues raised in the Weaknesses section.

---

> ### Author Response · Authors · 2025-11-27
> **Reply to Reviewer 2PbA**
>
> Thank you for your evaluation of our work! Below is our response to the seven concerns you raised in questions:
>
> ## Response to the 1st concern:
> Thanks for your great advice, we have incorporated the article you mentioned into the related work section of the revised version.
>
> ## Response to the 2nd concern:
>
> Typical curvature-aware downsampling(like proposed in Towards Scalable 3D Anomaly Detection and Localization: A Benchmark via 3D Anomaly Synthesis and A Self-Supervised Learning Network) involves patch division, curvature calculation, and the selection of top-k points. In our approach, we made slight modifications to the selection of top-k points. Specifically, we categorize points into three tiers based on their curvature ranking and discard points from each tier proportionally. This approach helps to avoid the risk of losing too much information from low-curvature regions, where anomaly can likely to appear.
>
> The improvements of PCGC over PointNet are discussed in the final paragraph of Section 3.3. PointNet requires the "Sampling—Grouping—PointNet" process at each feature aggregation step. In contrast, PCGC is based on the graph obtained during the CAS process, where each node is both defined and constrained. This eliminates the need for resampling or performing KNN searches at each step of aggregation, effectively adopting a GCN paradigm. Furthermore, the fusion strategy in PCGC is more complex compared to PointNet++, as it incorporates dimensional concatenation during token fusion, followed by downsampling through an MLP before pooling. Additionally, the introduction of FFN enhances the model's expressive power.
>
>
> ## Response to the 3rd & 5th concern:
>
> Actually, all PointMAE/PointBert-based baselines evaluated in this study, including Reg3AD, PatchCore (PointMAE), M3DM (PointMAE), and M3DM (PointBert), follow the segmentation-style setup of PointMAE, employing 12 Transformer blocks and extracting features from the 4th, 8th, and 12th blocks for anomaly detection. Likewise, MC3D-AD still requires 6 Point Transformer encoders and 6 Point Transformer decoders. In contrast, EG3AD achieves better performance using only 3 lightweight PCGC encoders and 3 Point Transformer decoders.
>
> We further tested the overall efficiency of EG3AD (which includes CAS, PCGC, and FPM in the encoder, as well as the decoder) in terms of Parameters, Inference Time, and FLOPs. This serves as a supplement to the ablation study on the complexity, computational cost, and performance of the PCGC encoder with varying numbers of PCGC blocks, which only considered the PCGC. The experimental setup is consistent with the tests on the Real3D-AD dataset. We set up three experiments:
> - Setting 1 corresponds to an anomaly detection model using the Point Transformer trained with PointMAE for inference, such as MC3DAD, Reg3DAD, M3DM (PointMAE), etc.;
> - Setting 2 corresponds to the result where the FPS in Setting 1 is replaced with CAS, to verify the impact of CAS on inference time efficiency;
> - Setting 3 corresponds to the performance of EG3AD.
>
> The experimental results are shown below:
>
>
> ### Table: Efficiency Test of Parameter, Inference Time, and FLOPs
>
> | **Setting** | **FLOPs** | **Throughput (samples/s)** | **Parameters (M)** |
> |-------------|-----------|----------------------------|--------------------|
> | **Setting 1** | 9.531G   | 65.88                      | 22.1M              |
> | **Setting 2** | 9.531G   | 48.47                      | 22.1M              |
> | **Setting 3** | 8.72G    | 47.4                       | 7.793M             |
>
> The encoder proposed by EG3AD enhances the encoding capability and reduces the dependence on Transformer blocks, leading to a significant improvement in parameter efficiency. The introduction of CAS requires solving PCA for each patch, which increases inference time. In terms of FLOPs, EG3AD shows some improvement compared to Point Transformer-based methods.
>
>
>
> ## Response to the 4th concern:
> Figure 4 indeed has an issue. During the plotting, we used too large a neighborhood to compute the normal vectors and applied Gaussian smoothing in order to highlight the curvature effect. However, this caused the curvature calculation to be insensitive to concave regions with small angles. We have now selected more appropriate parameters and will include the revised version in the submission.

---

> > ### Author Response · Authors · 2025-11-27
> > **Reply to Reviewer 2PbA(Continue)**
> >
> > ## Response to the 6th concern:
> > Additional visual comparisons will be organized and included in the appendix after the article is accepted.
> >
> > ## Response to the 7th concern:
> > The motivation behind our work on multi-class anomaly detection is to address the generalization problem present in previous single-class anomaly detection methods. This problem has been largely solved in 2DAD but remains a challenge in 3DAD. Our experimental results demonstrate that our method not only surpasses or matches the performance of single-class anomaly detection but also achieves leading stability across different classes.
> >
> > In accordance with the reviewer’s suggestion, we still conducted single-class experiments as well. In the Real3D-AD dataset, we fine-tuned the pre-trained model on point clouds from the same category and tested the results. The results are shown in the table below:
> >
> > ### Table: Performance Comparison between Multi-class and Single-class Settings
> >
> > | **Method**            | **Airplane** | **Car** | **Candybar** | **Chicken** | **Diamond** | **Duck** | **Fish** | **Gemstone** | **Seahorse** | **Shell** | **Starfish** | **Toffees** | **Avg.** |
> > |-----------------------|--------------|---------|--------------|-------------|-------------|----------|----------|--------------|--------------|-----------|--------------|-------------|----------|
> > | **Multi-class Setting** | 0.867        | 0.773   | 0.85         | 0.821       | 0.962       | 0.847    | 0.924    | 0.719        | 0.756        | 0.799     | 0.8          | 0.831       | 0.829    |
> > | **Single-class Setting** | 0.884        | 0.852   | 0.867        | 0.89        | 0.962       | 0.916    | 0.93     | 0.823        | 0.834        | 0.848     | 0.872        | 0.879       | 0.88     |
> >
> > If the above responses do not address your concerns, feel free to ask again!

---

> ### Author Response · Authors · 2025-12-01
> **Summary to My Rebuttal**
>
> This paper aims to propose an encoder specifically designed for multi-class anomaly detection tasks, EG3AD, with a focus on improving three encoding stages: tokenizing, shallow feature encoding, and deep feature encoding. To address these stages, we introduce CAS, PCGC, and FPM, respectively.
>
> The reviewers praised the leading performance of EG3AD in anomaly detection but questioned its efficiency. Through rebuttal experiments, we have confirmed to the reviewers that EG3AD saves a significant amount of parameters, highlighting its parameter efficiency.
>
> We have also corrected the issue raised by the reviewers regarding Figure 4 in the revised version.
>
> The reviewers' skepticism regarding the innovativeness of CAS was somehow unfounded. So we utilize previous works to clearly demonstrate the novelty of our approach.
>
> It is worth noting that the reviewers did not fully engage with the proposed method in the paper. This is evidenced by their request for clarification on the distinction between PCGC and PointNet++, despite the fact that we have provided a detailed description of the structure and implementation details of PCGC in the paper. A careful reading would have made the differences between PCGC and PointNet++ clearly apparent.
>
> Although we completed the additional experiments on single-class anomaly detection as requested by the reviewers, this experiment is exactly trivial. In fact, we have already demonstrated that our method achieves better results in terms of both accuracy and generalization when trained on datasets with multiple classes, compared to prior works trained on single-class datasets. This highlights the advancement of our method in both accuracy and generalization. The much effort spent on this experiment, while fulfilling the reviewers' request, did not contribute meaningfully to the core contributions of EG3AD.

---

### Official Review · Reviewer_VvwF · 2025-11-01

**Soundness:** 2
**Presentation:** 2
**Contribution:** 2
**Rating:** 4
**Confidence:** 4

**Summary:**

The paper tackles 3D point-cloud anomaly detection with three components: Curvature-Aware Sampling (CAS) to adaptively densify complex regions, Point Cluster Graph Convolution (PCGC) as a locality-aware, hierarchical encoder, and a Feature Purification Module (FPM) to denoise anomaly-corrupted features. On Real3D-AD and Anomaly-ShapeNet, EG3AD reports top object-level AUROC (e.g., ~81.1% on Real3D-AD; 85.6% on Anomaly-ShapeNet), with module ablations supporting each component.

**Strengths:**

1. Consistent performance gains are observed across both PCAD benchmarks. The per-module ablation studies provide useful insights.

2. The focus on efficiency—through locality using PCGC and sampling via CAS—is well justified, especially when dealing with large-scale point sets.

**Weaknesses:**

1. The discussion of deployment scenarios remains limited, with little coverage of sensor noise, sparsity, partial scans, or mixed categories. Comparisons to RGB-3D fusion-based anomaly detection methods are also lacking.

2. The experimental results appear to be based on a single random seed. There are no uncertainty estimates, statistical significance tests, or cross-dataset transfer analysis.

**Questions:**

1. Can results include variance over multiple runs and statistical significance tests for the reported AUROC scores?

2. How does the method perform under severe occlusions or partial scans? Is there any evaluation under domain shift, such as cross-dataset tests?

3. How does inference time compare to transformer-based baselines when operating on the same number of points?

---

> ### Author Response · Authors · 2025-11-27
> **Reply to Reviewer VvwF**
>
> to VvwF:
>
> Thank you for your evaluation of our work! Below is our response to the three concerns you raised in questions:
>
> ## Response to the 1st Concern:
> Following the reviewer’s suggestion, we conducted a random seed sensitivity experiment to evaluate the stability of our model with respect to random initialization. The training of EG3DAD consists of a pre-training stage and a fine-tuning stage. We perform the random seed experiments only in the fine-tuning stage. Specifically, under a fixed data split, we train the model with five different random seeds (42, 43, 44, 45, 46) and evaluate the AUROC on the Real3D-AD dataset at both the point level and the object level.
>
> Detailed results are provided below:
>
> ### Object-level AUROC on Real3D-AD (mean ± std over 5 random seeds)
>
> | **Method** | **Airplane** | **Car** | **Candybar** | **Chicken** | **Diamond** | **Duck** | **Fish** | **Gemstone** | **Seahorse** | **Shell** | **Starfish** | **Toffees** | **Avg.** |
> |------------|--------------|---------|--------------|-------------|-------------|----------|----------|--------------|--------------|-----------|--------------|-------------|----------|
> | **Ours**   | $0.867 \pm 0.024$ | $0.733 \pm 0.018$ | $0.850 \pm 0.007$ | $0.821 \pm 0.019$ | $0.962 \pm 0.007$ | $0.847 \pm 0.021$ | $0.924 \pm 0.023$ | $0.719 \pm 0.011$ | $0.767 \pm 0.026$ | $0.799 \pm 0.021$ | $0.800 \pm 0.005$ | $0.831 \pm 0.019$ | $0.830 \pm 0.007$ |
>
> ---
>
> ## Response to the 2nd Concern:
> In this work, the experiments follow prior related studies to ensure a fair comparison. In fact, we are not sure if sensor noise or partial occlusion can happen in real-world anomaly detection systems. However, existing point cloud anomaly detection datasets do not consider the scenarios you mentioned, making it difficult to conduct experiments for comparison with other benchmark methods.
>
> Regarding sensor noise, the two most widely used point cloud anomaly detection datasets, **Real3D-AD** and **MVTec3D-AD**, were scanned using high-precision instruments, and the sensor noise and point cloud scale can be considered negligible in comparison.
>
> To verify the impact of noise interference on EG3AD, we constructed a set of experiments. On the **Real3D-AD** dataset, we added Gaussian noise with scales of 1% and 5% to each point. The experiment demonstrates that Gaussian noise with a 1% scale standard deviation has a tolerable impact on performance, causing only a 0.03 AUROC drop, reflecting the robustness of EG3AD. However, Gaussian noise with a 5% scale standard deviation significantly degrades the model’s predictive ability, dropping by 16% in AUROC. This is because the flaw scale in PCAD is relatively small (as described in the supplementary materials on the Real3DAD dataset), with flaw points occupying less than 5% of the overall point cloud. Detailed results are provided below:
>
> ### Point Level AUROC Robustness to Noise of EG3AD on Real3D-AD
>
> | **Method**          | **Airplane** | **Car** | **Candybar** | **Chicken** | **Diamond** | **Duck** | **Fish** | **Gemstone** | **Seahorse** | **Shell** | **Starfish** | **Toffees** | **Avg.** |
> |---------------------|--------------|---------|--------------|-------------|-------------|----------|----------|--------------|--------------|-----------|--------------|-------------|----------|
> | **Ours**            | 0.763        | 0.825   | 0.881        | 0.700       | 0.938       | 0.844    | 0.902    | 0.534        | 0.731        | 0.790     | 0.753        | 0.862       | 0.794    |
> | **Ours + 1% Noise** | 0.748        | 0.772   | 0.819        | 0.683       | 0.851       | 0.800    | 0.836    | 0.530        | 0.707        | 0.756     | 0.728        | 0.815       | 0.760    |
> | **Ours + 5% Noise** | 0.630        | 0.593   | 0.709        | 0.592       | 0.626       | 0.683    | 0.719    | 0.528        | 0.583        | 0.640     | 0.617        | 0.631       | 0.629    |
>
> ---
>
> Regarding the **Cross-dataset** evaluation, we pre-trained on **Shapenet**, fine-tuned on **Anomaly Shapenet**, and then tested on **Real3DAD**. The results are as follows:
>
> ### Point Level AUROC on Cross-Dataset Evaluation
>
> | **Method**                  | **Airplane** | **Car** | **Candybar** | **Chicken** | **Diamond** | **Duck** | **Fish** | **Gemstone** | **Seahorse** | **Shell** | **Starfish** | **Toffees** | **Avg.** |
> |-----------------------------|--------------|---------|--------------|-------------|-------------|----------|----------|--------------|--------------|-----------|--------------|-------------|----------|
> | **Ours (Cross-dataset)**     | 0.553        | 0.661   | 0.733        | 0.522       | 0.634       | 0.687    | 0.665    | 0.397        | 0.624        | 0.598     | 0.545        | 0.642       | 0.605    |
> | **Ours (Oracle)**            | 0.763        | 0.825   | 0.881        | 0.7         | 0.938       | 0.844    | 0.902    | 0.534        | 0.731        | 0.79      | 0.753        | 0.862       | 0.794    |

---

> > ### Author Response · Authors · 2025-11-27
> > **Reply to Reviewer VvwF(Continue)**
> >
> > ## Response to the 3rd Concern:
> > Following your suggestion, we have included experiments on efficiency regarding parameters, FLOPs, and inference time. Setting 1 corresponds to an anomaly detection model using the Point Transformer trained with PointMAE for inference, such as MC3DAD, Reg3DAD, M3DM(PointMAE), etc.; Setting 2 corresponds to the result where the FPS in Setting 1 is replaced with CAS, to verify the impact of CAS on inference time efficiency; Setting 3 corresponds to the performance of EG3AD. The experiments show a significant improvement in parameter efficiency compared to prior work (from 22.1M to 7.793M). However, due to the introduction of the CAS, there is a slight decrease in inference efficiency, with throughput dropping from 65.88 samples/s to 47.4 samples/s. Detailed results are provided below:
> >
> > ### Efficiency Test of Parameter, Inference Time, and FLOPs
> >
> > | **Setting**  | **FLOPs**  | **Throughput (samples/s)** | **Parameters (M)** |
> > |--------------|------------|----------------------------|--------------------|
> > | **Setting 1** | 9.531G    | 65.88                      | 22.1M              |
> > | **Setting 2** | 9.531G    | 48.47                      | 22.1M              |
> > | **Setting 3** | 8.72G     | 47.4                       | 7.793M             |
> >
> > ---
> >
> > If the above responses do not address your concerns, feel free to ask again!

---

> ### Author Response · Authors · 2025-12-01
> **Summary to My Rebuttal**
>
> This paper aims to propose an encoder specifically designed for multi-class anomaly detection tasks, with a focus on improving three encoding stages: tokenizing, shallow feature encoding, and deep feature encoding. To address these stages, we introduce CAS, PCGC, and FPM, respectively.
>
> The reviewers praised the contributions of CAS and PCGC to the accuracy of anomaly detection. We have supplemented all the experiments suggested by the reviewers, demonstrating the statistical significance of our method, as well as verifying its robustness against sensor noise. While efficiency experiments indicate a slight increase in inference time, they demonstrate the significant parameter efficiency of our method.

---

### Official Review · Reviewer_whZ9 · 2025-11-08

**Soundness:** 2
**Presentation:** 3
**Contribution:** 2
**Rating:** 6
**Confidence:** 3

**Summary:**

This paper presents a novel anomaly detection approach for 3D point cloud data, aiming to develop a more efficient and effective solution to this challenging task. The proposed method comprises three key modules: CAS, PCGC, and FPM. Specifically, the CAS module enriches the geometric information embedded in the input tokens; the PCGC module refines local geometric features to enhance anomaly detection performance; and the FPM module employs optimal transport to extract informative features for reconstruction. Experimental results on the Real 3D-AD dataset substantiate the effectiveness of the proposed approach.

**Strengths:**

1. The paper proposes an efficient and effective method specifically designed for 3D point cloud anomaly detection, addressing a critical gap in this research domain.

2. The approach introduces the concept of optimal transport in a novel way to overcome the inherent challenges of 3D point cloud anomaly detection, representing a meaningful technical innovation.

3. The proposed method achieves SOTA performance on several classes of the Real 3D-AD dataset, highlighting its strong competitive advantage and empirical effectiveness.

**Weaknesses:**

1. The paper would benefit from a clearer and more comprehensive explanation of the PCGC module, including its internal architecture and the specific mechanisms that contribute to performance improvement.

2. The authors employ ShapeNet55 for pre-training and evaluate the model on the Anomaly ShapeNet dataset. Since the latter is synthesized from the former, there exists a potential risk of data leakage or overlap between the pre-training and test sets.

3. The proposed model requires a pre-training stage on ShapeNet55, followed by fine-tuning on the target dataset. Consequently, it utilizes substantially more data for training compared to the baseline methods. To make the comparison convincing, the authors are encouraged to include results against CLIP-based methods, such as PointAD, which adapts CLIP for point cloud anomaly detection. It would be valuable to investigate whether the proposed approach can still achieve superior performance when PointAD is evaluated under the same multi-class setting.

4. It is recommended to include additional supplementary experiments on the MVTec3D dataset to further validate the generalization ability of the proposed approach.

**Questions:**

See Weaknesses

---

> ### Author Response · Authors · 2025-11-27
> **Reply to Reviewer whZ9**
>
> Thank you for your positive evaluation of our work! Below we provide our responses to the four concerns you raised regarding the weaknesses of the paper.
>
> ---
>
> ## Response to the 1st Concern
>
> Our paper indeed provides comprehensive details about the Point Cloud Graph Convolution (PCGC) module, including its architecture (Figure 3(b), and the second and third paragraphs of Section 3.3), the mechanism underlying its performance improvement (last paragraph of Section 3.3), and the associated implementation hyperparameters (implementation details in section 4.0 and the appendix E.4.).
>
> The reasons for the observed performance improvement can be summarized as follows:
>
> - Following the paradigm of graph convolutions, the receptive field of the extracted features gradually expands from local to global. With relatively few encoder blocks, the model exhibits stronger inductive bias compared with the Transformer attention mechanism.
> - In comparison to PointNet++, PCGC performs additional feature fusion along the channel dimension after feature concatenation, thereby introducing more non-linearity into the representation.
> - Furthermore, the introduction of a feed-forward network (FFN) based on MLPs enhances the expressive capacity of the model.
>
> ---
>
> ## Response to the 2nd Concern
>
> We agree that there is a potential risk of data leakage. To investigate this issue, we designed the following experiment: we included the entire training set of Anomaly-Shapenet and the Real3D-AD dataset in the pretraining process. Afterward, we fine-tuned the model and evaluated its performance on these two datasets.
>
> For Anomaly-Shapenet, the prediction results remained unchanged (point-level AUROC = 0.878), which initially raised our suspicion. Surprisingly, the test performance on Real3D-AD also showed no significant change: the point-level AUROC only increased from 0.794 to 0.796, which is far below our expectation. We believe this is because, during pretraining, the model primarily learns generalizable representations of point clouds, rather than task-specific knowledge. The capability to reconstruct the normal form from anomalous samples is mainly acquired during the fine-tuning stage, and this reconstruction ability is precisely what is most crucial for anomaly detection.
>
> We in fact follow the same experimental setting as the official benchmark for the Anomaly-Shapenet dataset. For example, **IMRNet**, which is introduced in the same paper as Anomaly-Shapenet, is pretrained on ShapeNet55 (as indicated in their implementation details). Other classical methods in the Anomaly-Shapenet benchmark, such as **Reg3DAD**, **M3DM (PointMAE)**, **M3DM (PointBert)**, and **Patchcore (PointMAE)**, also rely on pretrained checkpoints provided by PointMAE or PointBert on GitHub. These checkpoints are first pretrained on ShapeNet and then fine-tuned on other downstream task datasets.
>
> ---
>
> ## Response to the 3rd Concern
>
> Most point cloud anomaly detection (PCAD) methods indeed require a pretraining stage. However, a key distinction between our work and prior approaches lies in how this stage is utilized. Previous works primarily build PCAD frameworks on top of pretrained Point Transformer architectures. Concretely, they directly load pretrained parameters from PointMAE/PointBert, obtained from their official GitHub repositories, and then fine-tune these models on the PCAD dataset.
>
> In contrast, our method discards the Point Transformer feature extractor and redesigns the feature extraction backbone to better match the characteristics of the anomaly detection task. Consequently, the pretraining process, including its design motivations and configurations, is explicitly discussed in our paper.
>
> CLIP-based methods such as PointAD constitute strong zero-shot baselines. However, our method and PointAD operate under finetuning paradigms. In fact, **EG3AD** achieves substantially higher detection accuracy than PointAD. On the Real3D-AD dataset, the average AUROC for object-level and point-level detection reported by PointAD is 0.748 and 0.735, respectively, whereas our method achieves 0.830 and 0.794.
>
> ### Detailed results on Real3D-AD (Point-level AUROC)
>
> | **Method**   | **Airplane** | **Car** | **Candybar** | **Chicken** | **Diamond** | **Duck** | **Fish** | **Gemstone** | **Seahorse** | **Shell** | **Starfish** | **Toffees** | **Avg.** |
> |--------------|--------------|---------|--------------|-------------|-------------|----------|----------|--------------|--------------|-----------|--------------|-------------|----------|
> | **PointAD**  | 0.672        | 0.723   | 0.713        | 0.677       | 0.877       | 0.510    | 0.801    | **0.802**    | **0.748**    | 0.778     | **0.814**    | 0.700       | 0.735    |
> | **Ours**     | **0.763**    | **0.825**| **0.881**    | **0.700**   | **0.938**   | **0.844** | **0.902** | 0.534        | 0.731        | **0.790** | 0.753        | **0.862**   | **0.794** |
>
> ---

---

> > ### Author Response · Authors · 2025-11-27
> > **Reply to Reviewer whZ9(Continue)**
> >
> > ## Response to the 4th Concern
> >
> > The performance of EG3AD on the MVTec 3D-AD dataset is reported in Table below. We keep the experimental settings consistent with those used for Real3D-AD. Since MVTec 3D-AD is a multimodal dataset containing both RGB images and single-view point cloud scans, and EG3AD operates solely on point cloud data, the multimodal baselines (e.g., M3DM) are evaluated using only the point cloud modality to ensure a fair comparison.
> >
> > ### MVTec-3D object-level AUROC results
> >
> > | **Method**   | **Bagel**  | **Cable Gland** | **Carrot** | **Cookie** | **Dowel** | **Foam**  | **Peach** | **Potato** | **Rope**  | **Tire**  | **Mean**  |
> > |--------------|------------|-----------------|------------|------------|-----------|-----------|-----------|------------|-----------|-----------|-----------|
> > | **3D-ST**    | 0.862      | 0.484           | 0.832      | 0.894      | 0.848     | 0.663     | 0.763     | 0.687      | **0.958** | 0.486     | 0.748     |
> > | **FPFH**     | 0.825      | 0.551           | 0.952      | 0.797      | 0.883     | 0.582     | 0.758     | 0.889      | 0.929     | 0.653     | 0.782     |
> > | **AST**      | 0.881      | 0.576           | 0.965  | 0.957  | 0.679     | 0.797     | **0.990** | 0.915      | 0.956     | 0.611     | 0.833     |
> > | **M3DM**     | 0.941  | 0.651           | 0.965  | 0.969      | 0.905 | 0.760     | 0.880     | **0.974**  | 0.926     | 0.765     | 0.874     |
> > | **Group3AD** | **0.989**  | 0.722           | 0.976  | 0.954      | 0.852     | 0.769     | 0.910     | 0.940      | 0.909     | **0.864** | 0.889     |
> > | **Ours**     | 0.973      | **0.768**       | **0.982**      | **0.972**      | **0.934**     | **0.810**     | 0.939     | 0.955      | 0.899     | 0.819     | **0.905** |
> >
> >
> > ---
> >
> > If the above responses do not fully address your concerns, please feel free to let us know.

---

> ### Author Response · Authors · 2025-12-01
> **Summary of My Rebuttal**
>
> This paper aims to propose an encoder specifically designed for multi-class anomaly detection tasks, with a focus on improving three stages of encoding: tokenizing, shallow feature encoding, and deep feature encoding. In terms of tokenizing, we introduce CAS, which enables better representation of geometric details in point clouds with the same number of points. For shallow feature encoding, we design PCGC, which leverages inductive priors to efficiently extract local features with fewer parameters. In terms of deep feature encoding, we propose FPM, which uses clustering based on optimal transport to perform deep feature extraction.
>
> The reviewers have acknowledged the leading performance of our detection method, and in the rebuttal additional experiments, we demonstrate a significant improvement in parameter efficiency.
>
> We have addressed all the questions raised by the reviewers and experimentally verified previously unaddressed issues—specifically, the extent of data leakage caused by using models pre-trained on ShapeNet and subsequently tested on Anomaly ShapeNet for anomaly detection. Supplementary experiments on MVTec3DAD further highlight the generalizability of our method and its practical applicability for anomaly detection in industrial scenarios.

---

### Comment · Area_Chair_swFA · 2025-11-26
**Important Ethics Concern: Proper Attribution Required for Figures**

Dear Authors of Submission3215,

As flagged by Reviewer 22cG, your appendix contains two figures (Figures 6 and 7) that are directly taken from previously published papers [1, 2] without explicit attribution or clarification. To avoid any further ethics concerns and potential involvement of the ICLR 2026 Ethics Chairs, please revise your manuscript to properly cite the relevant papers and clearly indicate the sources of these figures.

[1] Real3D-AD: A Dataset of Point Cloud Anomaly Detection. NeurIPS 2023.
[2] Towards Scalable 3D Anomaly Detection and Localization: A Benchmark via 3D Anomaly Synthesis and A Self-Supervised Learning Network. CVPR 2024.

Thanks for your attention to this matter.

Best regards,
AC

---

> ### Author Response · Authors · 2025-11-27
> **Reply to Area Chair**
>
> Dear Area Chair,
>
> We have noted the ethics concern raised by Reviewer 22cG, and we appreciate your feedback. In response, we had included figures and tables from the dataset’s GitHub repository in the appendix to provide a clear introduction to the dataset used. However, to ensure compliance with ICLR’s guidelines, we have decided to remove the related content (Figure 6, Figures 7, and Table 7). A revised version will be uploaded.
>
> Best regards,
> Author

---

### Meta-Review · Area_Chair_ZLL6 · 2026-01-01

**Summary:**

Four reviewers assessed this paper, resulting in a consensus for rejection with three reviewers initially assigning weak reject ratings and one leaning towards weak acceptance. While the authors proposed an efficient 3D anomaly detection framework, the review process uncovered fundamental methodological and experimental flaws that were not adequately resolved during the rebuttal. The primary grounds for rejection include a significant risk of data leakage due to overlaps between the pre-training and test sets, questionable novelty regarding the CAS module's similarity to existing curvature-based downsampling, and insufficient comparative analysis against state-of-the-art methods in terms of efficiency and RGB-3D fusion. Furthermore, the method demonstrated poor robustness to noise and relied on sensitive heuristics, limiting its practical applicability. Consequently, the AC recommends rejection.

**Reviewer Concerns:**

While the authors successfully addressed requests for additional experiments on the MVTec3D dataset, comparisons with PointAD (CLIP-based method), and statistical rigor regarding seed variance, several critical concerns remain unresolved. A paramount issue raised by Reviewers whZ9 and 22cG is the potential data leakage arising from pre-training on ShapeNet55 while testing on Anomaly-ShapeNet; the rebuttal failed to eliminate this concern, which fundamentally compromises the validity of the reported performance. Regarding novelty, Reviewer 2PbA’s critique that the CAS module lacks distinction from standard curvature-based downsampling was not convincingly refuted, nor was the theoretical link between curvature estimation and anomaly detection established. Efficiency claims challenged by Reviewer 22cG remain unsubstantiated; although the authors provided parameter and FLOP counts, they failed to offer a holistic comparison against SOTA methods. Finally, the method's robustness is questionable, as additional experiments revealed significant performance degradation under sensor noise (Reviewer VvwF) and high sensitivity to heuristic hyperparameters (Reviewer 22cG), indicating limited utility in realistic deployment scenarios.

**Reviewer Scores:**

Reviewer whZ9 would likely lower their score to a 4, as the rebuttal failed to address the critical issue of data leakage, which undermines the positive aspects of the evaluation that initially prompted the weak accept rating. Reviewer VvwF is expected to maintain their score of 4, given that the requested comparisons to RGB-3D fusion methods were ignored and the new noise sensitivity experiments confirmed the reviewer's suspicions about deployment fragility. Reviewer 2PbA would also likely retain a score of 4, or potentially lower it, because the authors provided unconvincing arguments regarding the novelty of the CAS module and the realism of the visualizations, leaving the core theoretical contribution in doubt. Reviewer 22cG is expected to keep their score of 4; the authors did not provide the requested comparative efficiency analysis against other methods and failed to address the overfitting risks associated with the heuristic hyperparameters and the data leakage issue.

---

### Decision · Program_Chairs · 2026-01-26

Reject